# Unstructured regions in IRE1α specify BiP-mediated destabilisation of the luminal domain dimer and repression of the UPR

Niko Amin-Wetzel[1†‡], Lisa Neidhardt[1†], Yahui Yan[1], Matthias P Mayer[2], David Ron[1]*

[1]Cambridge Institute for Medical Research (CIMR), University of Cambridge, Cambridge, United Kingdom; [2]Center for Molecular Biology of Heidelberg University (ZMBH), DKFZ-ZMBH Alliance, Heidelberg, Germany

**Abstract** Coupling of endoplasmic reticulum (ER) stress to dimerisation-dependent activation of the UPR transducer IRE1 is incompletely understood. Whilst the luminal co-chaperone ERdj4 promotes a complex between the Hsp70 BiP and IRE1's stress-sensing luminal domain (IRE1$^{LD}$) that favours the latter's monomeric inactive state and loss of ERdj4 de-represses IRE1, evidence linking these cellular and in vitro observations is presently lacking. We report that enforced loading of endogenous BiP onto endogenous IRE1α repressed UPR signalling in CHO cells and deletions in the IRE1α locus that de-repressed the UPR in cells, encode flexible regions of IRE1$^{LD}$ that mediated BiP-induced monomerisation in vitro. Changes in the hydrogen exchange mass spectrometry profile of IRE1$^{LD}$ induced by ERdj4 and BiP confirmed monomerisation and were consistent with active destabilisation of the IRE1$^{LD}$ dimer. Together, these observations support a competition model whereby waning ER stress passively partitions ERdj4 and BiP to IRE1$^{LD}$ to initiate active repression of UPR signalling.

*For correspondence:
dr360@medschl.cam.ac.uk

[†]These authors contributed equally to this work

Present address: [‡]Institute of Science and Technology Austria (IST Austria), Klosterneuburg, Austria

## Introduction

In eukaryotes, the endoplasmic reticulum (ER) is the central organelle for the synthesis of proteins destined for secretion and membrane insertion. The ER lumen harbours a specialised protein folding and processing machinery that constitutes the protein folding capacity of the ER. To ensure that the environment for productive protein maturation is maintained, both folding capacity and the inward flux of newly synthesised proteins are regulated by a pervasive negative feedback signalling pathway, the unfolded protein response (UPR) (*Kozutsumi et al., 1988*; *Cox et al., 1997*). In mammalian cells, this pathway involves three known signaling branches each directed by a unique signal transducer resident in the ER membrane, IRE1, PERK and ATF6. An imbalance between folding load and capacity (ER stress) activates these sensors initiating a rectifying transcriptional and translational response to defend protein-folding homeostasis in the compartment (reviewed in *Walter and Ron, 2011*). While details of downstream events and their physiological significance are relatively well characterised (reviewed in *Wang and Kaufman, 2016*), the molecular mechanisms of the earliest events in UPR activation remain incompletely understood.

IRE1, conserved in all eukaryotes and therefore the best-studied UPR transducer (*Cox et al., 1993*; *Mori et al., 1993*), detects ER stress via its luminal domain (IRE1$^{LD}$), initiating dimerisation-dependent autophosphorylation of its cytosolic domain (*Shamu and Walter, 1996*). The subsequent allosteric activation of the cytosolic endoribonuclease domain (*Lee et al., 2008*) leads to unconventional splicing of the mRNA encoding the XBP1/HAC1 transcription factor (*Cox and Walter, 1996*; *Yoshida et al., 2001*; *Calfon et al., 2002*), thereby promoting translation of an effector that drives a conserved gene-expression program.

**eLife digest** Cells produce many protein molecules. These are made of chains of building blocks called amino acids that then fold into three-dimensional shapes. Specialist proteins known as chaperones assist this folding process. For example, the chaperone BiP helps other proteins fold in a compartment within the cell called the endoplasmic reticulum.

To match the supply of chaperones to the demand of unfolded proteins, cells have stress receptors, such as IRE1 in the endoplasmic reticulum. IRE1 responds to changing levels of unfolded proteins by generating signals that tell cells whether they need more chaperones. Previous studies in a test tube suggest that when levels of unfolded proteins are low, BiP represses IRE1 signalling. However, when the levels of unfolded proteins increase, the unfolded proteins compete with IRE1 for BiP, releasing the brake BiP imposes on IRE1 signalling. It remained unclear if BiP regulates IRE1 in the same way in living cells.

To address this question, Amin-Wetzel, Neidhardt et al. studied IRE1 signalling in mammalian cells grown in the laboratory. The experiments revealed that cells containing a modified version of IRE1 to which BiP binds more strongly had less IRE1 signalling. On the other hand, cells containing versions of IRE1 that BiP binds less well had more active IRE1 signalling. These findings suggest that in cells, as in the test tube, unfolded proteins and IRE1 compete for BiP binding. This relationship comprises a simple mechanism allowing cells to sense and respond to the burden of unfolded proteins in their endoplasmic reticulum.

Over time, the amount of unfolded proteins in the cell likely contributes to the development of aging-related diseases such as adult-onset diabetes. A better understanding of how cells handle unfolded proteins may lead to more effective treatments for these diseases.

Two models have been put forth to describe how IRE1$^{LD}$ senses ER stress. A direct binding model posits that unfolded proteins act as ligands stabilising IRE1's dimeric/oligomeric state thereby promoting its activation. This model is supported by the crystal structure of the core luminal domain from *S. cerevisae* IRE1, showing an IRE1$^{LD}$ dimer interface traversed by a groove with architectural similarity to the major histocompatibility peptide-binding complexes (MHCs) (*Credle et al., 2005*). Peptide ligands of the yeast IRE1$^{LD}$ have been identified and their addition to dilute solutions of yeast IRE1$^{LD}$ enhances the population of higher order species, although a clear shift from monomers to dimers was not readily observable (*Gardner and Walter, 2011*).

The luminal domain of the broadly expressed alpha isoform of human IRE1 (hIRE1$\alpha^{LD}$) also crystallises as a dimer, with an overall architecture similar to the yeast protein, however, barring conformational changes, the MHC-like groove is too narrow to accommodate a peptide (*Zhou et al., 2006*). Recently, peptides have been identified that bind hIRE1$^{LD}$ and affect its oligomeric state, as assessed by analytical ultracentrifugation (AUC). Moreover, nuclear magnetic resonance (NMR) reported on peptide-induced structural rearrangements within the hIRE1$\alpha^{LD}$ that also affected residues near the MHC-like groove. Hence, it has been proposed that the structure of *Zhou et al. (2006)* represents a 'closed' conformation of the peptide-binding groove that can shift towards an 'open' state to allow peptide binding (*Karagöz et al., 2017*). However, a co-crystal structure of the ligand-bound yeast or human IRE1$^{LD}$ is not available and it remains unclear if and how peptide ligands affect hIRE1$^{LD}$ dimerisation, the first crucial step of its activation.

An alternative hypothesis posits that IRE1 is repressed by interacting with a major component of the ER folding machinery, the heat-shock protein (Hsp70) chaperone BiP. It is proposed that upon stress, unfolded proteins accumulate and compete for BiP interaction, thereby kinetically disrupting the inhibitory IRE1-BiP complex. This chaperone inhibition model draws parallels between the regulation of the UPR and its cytosolic counterpart, the heat-shock response, in which chaperones associate with the transcription factor Hsf1, in eukaryotes, and $\sigma^{32}$, in bacteria, to interfere with their activity (*Abravaya et al., 1992*; *Shi et al., 1998*; *Tomoyasu et al., 1998*). This model is supported by an inverse correlation between ER stress-induced IRE1 activity and the amount of ER-localised BiP recovered in complex with it (*Bertolotti et al., 2000*; *Okamura et al., 2000*; *Oikawa et al., 2009*).

Further molecular insight into the chaperone inhibition mechanism was gained recently by the discovery of ERdj4 as an ER-localised J-domain protein that selectively represses IRE1 activity in vivo and loads BiP onto the IRE1$^{LD}$, thereby promoting monomerisation in vitro (*Amin-Wetzel et al., 2017*). Whilst other modes of BiP binding to the IRE1$^{LD}$ have been proposed (*Carrara et al., 2015*; *Kopp et al., 2018*) the aforementioned observations suggest a mechanism in which BiP engages the IRE1$^{LD}$ as an Hsp70 substrate: ATP-bound BiP initially interacts with the IRE1$^{LD}$ with high $k_{on}$ and high $k_{off}$ rates and only captures IRE1$^{LD}$ as a substrate (in the ADP bound state, with low $k_{off}$ rates) after ERdj4 co-chaperone-instructed ATP hydrolysis. This model draws on the conventional view whereby J-domain proteins act as adaptors that enable efficient substrate recognition via their divergent targeting domains and subsequent binding of Hsp70s, promoted by their conserved J-domain that stimulates Hsp70's ATPase activity (reviewed in *Kampinga and Craig, 2010*). J-domain co-chaperones act in concert with nucleotide exchange factors (NEFs, reviewed in *Behnke et al., 2015*) to accelerate Hsp70s' cycles of substrate binding and release, resulting in substrate-selective ultra-affinity (*Misselwitz et al., 1998*; *De Los Rios and Barducci, 2014*), which is the basis for the assembly of Hsp70-substrate complexes.

Whilst ERdj4's repressive action on IRE1 signalling in cells and its ability to promote a complex between IRE1$^{LD}$ and BiP that favours the former's monomeric state in vitro fit the chaperone inhibition model, they remain correlative findings and may be causally unrelated. For example, it is possible that ERdj4's repressive action in cells arises from its role in eliminating IRE1$^{LD}$ activating ligands and not from catalysing the repressed, monomeric IRE1$^{LD}$-BiP complex observed in vitro. Here, in support of the chaperone inhibition model, we report that enforced targeting of endogenous BiP to endogenously-expressed IRE1$^{LD}$ represses UPR signalling in cells, thereby establishing that BiP can directly repress IRE1 in vivo and that features of the IRE1$^{LD}$ that specify its repression in cells also specify its ability to undergo actively-driven monomerisation by ERdj4 and BiP in vitro.

## Results

### BiP binding to IRE1$^{LD}$ represses IRE1 activity in cells

An inverse correlation between ER stress-induced IRE1 activity and the amount of BiP recovered in complex with it has been previously observed (*Bertolotti et al., 2000*; *Okamura et al., 2000*; *Oikawa et al., 2009*) but a causal link between BiP binding and IRE1 activity status had never been conclusively established. To assess the effect of BiP binding on the activity of IRE1 in vivo, we modified the endogenous *Ern1* locus to encode an ER targeted J-IRE1 fusion protein consisting of IRE1α's endogenous signal peptide, an N-terminally fused J-domain (derived from ERdj4) followed by the endogenous IRE1α coding sequence (*Figure 1—figure supplement 1*). The alpha isoform accounts for all measurable activity in CHO cells and is referred to as IRE1 hereafter. By employing this fusion protein, we expected to stimulate BiP's ATPase activity in close proximity to the IRE1$^{LD}$ thereby promoting formation of an IRE1-BiP complex. As control, a point mutant ERdj4 J-domain was used that had the histidine of the highly conserved HPD motif replaced by glutamine (J$^{QPD}$) compromising the stimulation of BiP's ATPase activity (*Wall et al., 1994*). The glycine-phenylalanine-rich (G/F) region of ERdj4 was included as a flexible linker, to allow the J-domain to explore the entire surface of IRE1$^{LD}$. We deemed that low level expression of endogenous IRE1 (and hence J-IRE1) would minimise IRE1-independent effects of this chimeric J-domain protein on the ER folding environment, effects that could not be excluded as having contributed to the previously-noted repressive effect of ERdj4 over-expression on the UPR (*Amin-Wetzel et al., 2017*).

Using an *Ern1* null cell line with a genomic deletion encompassing the IRE1$^{LD}$-encoding exons 2–12 (ΔIRE1, previously described in *Kono et al., 2017*), we reconstituted the endogenous locus with either wild-type IRE1, J-IRE1 or J$^{QPD}$-IRE1 fusion. Additionally, the cell lines stably expressed XBP1s::Turquoise and CHOP::GFP reporters that are controlled by the IRE1 and PERK UPR branches, respectively. Flow cytometry analysis showed that reconstitution of the locus with wild-type IRE1 rescues the non-responsive XBP1s::Turquoise phenotype of the ΔIRE1 cells towards stress induced by tunicamycin (*Figure 1A*). In comparison, cells expressing the J-IRE1 fusion showed low XBP1::Turquoise reporter levels, indicating repressed IRE1 activity, even under stress. Repression was dependent on the integrity of the J-domain as ΔIRE1 cells reconstituted with the mutant J$^{QPD}$-IRE1 acquired nearly wild-type stress responsiveness. The J-IRE1 protein was not otherwise compromised,

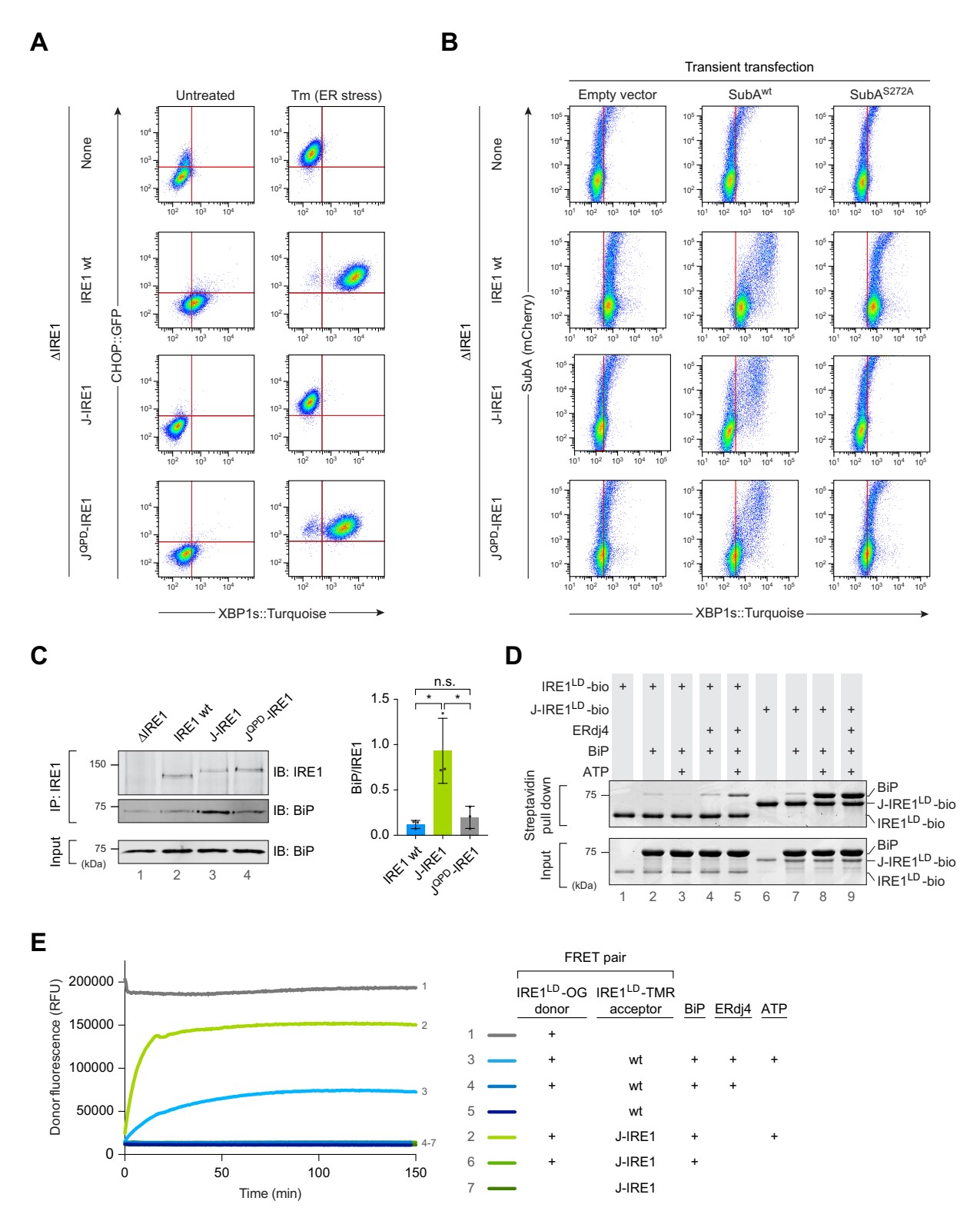

**Figure 1.** Fusion of ERdj4's J-domain to IRE1$^{LD}$ promotes efficient BiP association thereby repressing IRE1 activity in cells. (**A**) Two dimensional plots of CHOP::GFP and XBP1s::Turquoise signals from CHO-K1 dual UPR reporter cells stably expressing the indicated IRE1 variants [IRE1 wild-type (wt), J-IRE1 or J$^{QPD}$-IRE1 fusion; see *Figure 1—figure supplement 1A*, for schema of the alleles] from the endogenous *Ern1* locus untreated and treated with the ER stressor tunicamycin (Tm). Clones used for the analysis were derived from an IRE1 null (ΔIRE1) parental cell line. A representative data set out of *Figure 1 continued on next page*

*Figure 1 continued*

three independent experiments is shown. Note the low XBP1::Turquoise intensity in stressed J-IRE1 rescued ΔIRE1 cells. (B) Two dimensional plots of mCherry and XBP1s::Turquoise signals of clones described in 'A' transiently transfected with a plasmid encoding the SubA protease, which cleaves BiP at its interdomain linker and an mCherry fluorescent transfection marker. The inactive SubA$^{S272A}$ mutant was used as control. Representative data from nine biological repeats is shown. (C) Immunoblot (IB) of endogenous IRE1 and associated BiP recovered from the indicated cell lines by immunoprecipitation (IP) of IRE1. Quantification of the ratio of BiP to IRE1 signals in three independent experiments is shown on the right (mean ± standard deviation, n.s.: not significant, *: p<0.05, unpaired parametric Student's t test). BiP in input cell lysates is provided as loading control. (*Figure 1—source data 1*) (D) Coomassie-stained SDS-PAGE gel of biotinylated IRE1$^{LD}$ (IRE1$^{LD}$-bio) and a fusion of ERdj4's J-domain to IRE1$^{LD}$ (J-IRE1$^{LD}$-bio, as in 'A') and BiP, both recovered on a streptavidin matrix from samples constituted as indicated. Protein concentrations were 5 µM IRE1$^{LD}$-bio variants, 30 µM BiP, 8 µM ERdj4, and 2 mM ATP. Proteins were eluted in SDS sample buffer. A representative data set out of three independent experiments is shown. (E) Time-dependent change in donor fluorescence of the indicated IRE1$^{LD}$ FRET pair incubated at t = 0 with the components shown to the right. IRE1$^{LD}$ proteins were either labelled with the donor molecule Oregon green 488 (OG) or the acceptor molecule TAMRA (TMR). Protein concentrations were 0.2 µM IRE1$^{LD}$ FRET pair, 30 µM BiP, 2.5 µM ERdj4 and 2 mM ATP. A representative graph of three independent experiments is shown (*Figure 1—source data 2*).

The online version of this article includes the following source data and figure supplement(s) for figure 1:

**Source data 1.** Source data for *Figure 1C*.
**Source data 2.** Source data for *Figure 1E* .
**Figure supplement 1.** Modification of the endogenous *Ern1* locus to introduce IRE1$^{LD}$ variants.

as it was still able to respond to the ER stressor SubA, a protease that inactivates BiP by cleaving its interdomain linker (*Paton et al., 2006*) (*Figure 1B*). These findings are consistent with BiP serving as a direct *trans*-acting factor to specify repression mediated by a J-domain presented in cis to the IRE1$^{LD}$.

A role for the *cis*-active J-domain in recruiting BiP to the IRE1$^{LD}$ is supported by immunoprecipitation (IP) of endogenous IRE1 prepared from the cells described above. More BiP was recovered in complex with the J-IRE1 chimera compared to the wild-type IRE1 whilst the mutant J$^{QPD}$-IRE1 fusion associated with a similar amount of BiP as the wild-type (*Figure 1C*), which is in accordance to their similar phenotype detected by flow cytometry.

To further validate these in vivo observations, we reconstituted the system in vitro using recombinant proteins purified from bacteria. First, pull down of either C-terminally biotinylated IRE1$^{LD}$ (IRE1$^{LD}$-bio) or J-IRE1$^{LD}$ (J-IRE1$^{LD}$-bio) was performed. We assessed the formation of a BiP-IRE1$^{LD}$-bio complex on SDS-PAGE after recovery on immobilised streptavidin (*Figure 1D*). Whilst BiP recovery in complex with IRE1$^{LD}$-bio was dependent on the presence of both ERdj4 and ATP in the binding assay, complex formation of BiP and J-IRE1$^{LD}$-bio required only ATP.

Next, we tested how BiP binding affected J-IRE1$^{LD}$'s oligomeric status in vitro using a Förster resonance energy transfer (FRET)-based assay to continuously monitor the monomer-dimer equilibrium (as described previously, *Amin-Wetzel et al., 2017*). A donor IRE1$^{LD}$ labelled with Oregon Green (OG) was pre-equilibrated either with an IRE1$^{LD}$ or J-IRE1$^{LD}$ acceptor molecule labelled with TAMRA (TMR). As previously observed, BiP, ERdj4, and ATP were all required to monomerise the IRE1$^{LD}$ homodimer as reflected in the time-dependent increase in donor fluorescence until a kinetically maintained pseudo steady state was reached (*Figure 1E*). In contrast, heterodimeric FRET pairs containing the J-IRE1$^{LD}$ fusion and IRE1$^{LD}$ were monomerised by BiP in an ATP-dependent manner, but did not require ERdj4 in trans. The nucleotide-dependent, BiP-induced monomerisation of the J-IRE1$^{LD}$ containing heterodimer occurred with an approximately four-fold higher initial velocity and a higher plateau in the pseudo steady state of the reaction. Taken together these findings suggest that the fused J-domain enables efficient formation of the IRE1$^{LD}$-BiP complex, thereby promoting monomerisation, which leads to repression of IRE1 activity.

## In vitro characterisation of direct binding of unfolded proteins to IRE1$^{LD}$ as modelled by the MPZ-N peptide

To examine the role of peptides in regulating the monomer-dimer equilibrium of IRE1$^{LD}$ and hence its activity, we turned to a 12-mer peptide (MPZ-N) derived from myelin protein zero. MPZ-N is the best studied ligand for mammalian IRE1$^{LD}$ and was recently proposed to directly interact with the peptide-binding groove thereby influencing IRE1$^{LD}$'s oligomeric status (*Karagöz et al., 2017*). When introduced into the FRET-based assay, MPZ-N had no measurable effect on donor fluorescence.

However, as the optical readout of this assay is sensitive mostly to monomerisation (as reflected in an increase in donor fluorescence, *Figure 1E*) it would be a relatively insensitive measure of MPZ-N peptide driven dimerisation. Therefore, we sought different assays to report on the ability of the MPZ-N peptide to promote IRE1$^{LD}$ dimers.

The distribution of IRE1$^{LD}$ between monomers and dimers can be tracked by size exclusion chromatography (SEC), as evidenced by the concentration-dependence of the peak elution time of IRE1$^{LD}$ and two dimerisation-compromised mutants: a previously characterised W125A variant (*Zhou et al., 2006*) and a new, more severe P108A variant (*Figure 2—figure supplement 1A*). Both mutations are predicted to decrease hydrophobic interactions across the dimer interface (*Figure 2—figure supplement 1B*). Addition of MPZ-N peptide (at concentrations exceeding the reported K$_{1/2 \text{ max}}$ for binding of 16 µM, *Karagöz et al., 2017*) did not affect the peak elution time of IRE1$^{LD}$, itself introduced into the assay at 500 nM, near the K$_d$ for IRE1$^{LD}$ dimerisation (*Zhou et al., 2006*) (*Figure 2A and B*).

To confirm these observations, we made use of an alternative assay reporting on IRE1$^{LD}$'s dimerisation status. To this end, we employed a modified IRE1$^{LD \, Q105C}$ that forms a disulphide across the dimer interface, creating a covalently stabilised dimer when placed in oxidising conditions (*Figure 2—figure supplement 1C*) and the aforementioned dimerisation-compromised versions of the IRE1$^{LD}$ (W125A and P108A). Differential scanning fluorimetry (DSF) revealed that the melting temperature (T$_m$) of disulphide-linked IRE1$^{LD \, Q105C \, SS}$ was ~10 ˚C higher than the T$_m$ the wild-type protein, a T$_m$ difference that was effaced by reduction of the dimer-stabilising disulphide (*Figure 2—figure supplement 1D*). By contrast, the IRE1$^{LD}$ monomeric variants exhibited a T$_m$ 5–10 ˚C lower than the wild-type. These observations established a correlation between the monomer-dimer equilibrium and the T$_m$ of the protein consistent with dimerisation-mediated stabilisation of the IRE1$^{LD}$. A ligand, stabilising the IRE1$^{LD}$ dimer, is predicted to increase the T$_m$, however, addition of MPZ-N peptide had no effect on the T$_m$ of IRE1$^{LD}$ (*Figure 2—figure supplement 1D*, the significance of the lowering of T$_m$ observed at the highest concentrations of peptide remains to be determined).

To gain insight into the mode of MPZ-N binding to IRE1$^{LD}$ we made further use of the disulphide-linked IRE1$^{LD \, Q105C \, SS}$. The crystallised IRE1$^{LD \, Q105C \, SS}$ dimer proved identical in structure to the wild-type protein (root-mean squared deviation (RMSD) of 0.46 Å over 227 C$^{\alpha}$ atoms) except for the presence of a conspicuous density corresponding to a C105-C105 trans-protomer disulphide, thereby locking the proposed binding groove in the 'closed' conformation (*Figure 2C*, *Figure 2—figure supplement 1E* and *Table 1*). Nonetheless, a fluorescence polarisation binding assay, using FAM-labelled MPZ-N, showed that binding to the IRE1$^{LD}$ was not compromised by the disulphide (*Figure 2D*), leading us to conclude that MPZ-N does not obligatorily bind within the proposed MHC-like groove of the IRE1$^{LD}$. This conclusion is also consistent with the paramagnetic relaxation enhancement (PRE) experiments with IRE1$^{LD}$ and an MPZ-proxyl-labelled peptide (*Karagöz et al., 2017*), which present a distance constraint of 10 Å between Ile186 of the IRE1$^{LD}$ and the labelled Cys5 of the peptide. *Figure 2—figure supplement 2* shows that the extended peptide is free to explore the entire surface of one face of the IRE1$^{LD}$ and may therefore bind in locations other than the MHC-like groove, without violating this distance constraint.

## Identification of regions in IRE1$^{LD}$ involved in BiP-mediated regulation of its activity

Given the evidence for BiP's role in IRE1 repression, we tried to identify regions in IRE1$^{LD}$ that might be important for such regulation. BiP, as an Hsp70 chaperone, typically interacts with unfolded or flexible regions in its client proteins (*Rüdiger et al., 1997*) and we held that this might also be the case for its interaction with the IRE1$^{LD}$. Therefore, we sought clues to map these flexible regions by collecting data on the structural dynamics of IRE1$^{LD}$ in solution as evaluated by hydrogen-$^{1}$H/$^{2}$H-exchange experiments in combination with mass spectrometry (HX-MS).

IRE1$^{LD}$ was pre-equilibrated for 30 min at 30˚C followed by an exchange reaction in deuterium oxide (D$_2$O) buffer for 30 and 300 s. Subsequent analysis of deuteron incorporation was performed as described previously (*Hentze and Mayer, 2013*). Information on peptic peptides covering 85% of the IRE1$^{LD}$ sequence was obtained (*Table 2*). The extracted percentage of exchange (%ex) for each peptic peptide contained information about the thermodynamic stability of structural elements, the hydrogen bonding and solvent accessibility of backbone amide hydrogens (*Figure 3A* left panel). Projection of these values onto the crystal structure showed that regions in the hydrophobic core

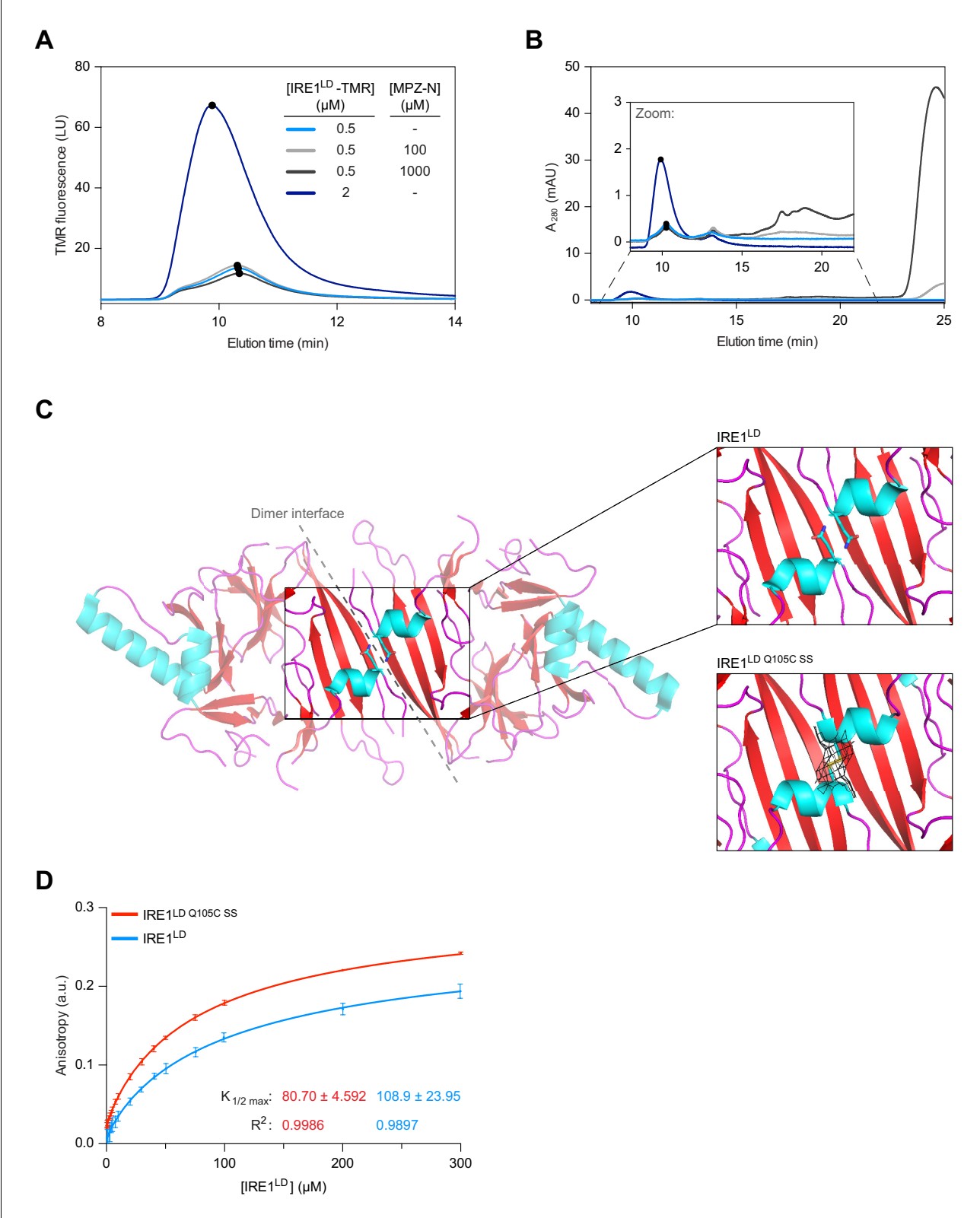

**Figure 2.** Binding of MPZ-N peptide to IRE1$^{LD}$ does not promote IRE1$^{LD}$ dimerisation. (A) Size-exclusion chromatography (SEC) elution profiles of TAMRA (TMR)-labelled wild-type IRE1$^{LD}$ at the indicated concentrations in presence and absence of MPZ-N peptide. TMR fluorescence is plotted against elution time (see: *Figure 2—source data 1*). (B) SEC elution profiles (as in 'A'), but with protein absorbance at 280 nm (A$_{280}$) plotted against elution time. The inset is a zoom into the segment of the chromatogram encompassing the IRE1$^{LD}$ proteins, whose absorbance is dwarfed by the peak

*Figure 2 continued on next page*

*Figure 2 continued*

of free peptide (eluting at ~24 min). The heterogenous peaks eluting between 15 and 20 min in the sample loaded with 1 mM MPZ-N peptide, likely reflected peptide oligomerisation (see: *Figure 2—source data 1*). (C) Cartoon representation of the IRE1$^{LD}$ dimer (PDB: 2HZ6) is shown on the left with coloured secondary structures (cyan for helices, red for sheets and magenta for loops). The Gln105 side chain is shown as sticks, a closer view of which is shown on the top right. The bottom right panel shows a similar view of the Gln105Cys mutant (crystallised here), which forms a disulphide bond, covered with clear electron density (black mesh represents the 2mFo − DFc map, contoured at 1.0 σ, including density within 2 Å of the cysteine residues). (D) Anisotropy of FAM labelled MPZ-N peptide (100 nM) in presence of increasing concentrations of either wild-type IRE1$^{LD}$ or disulphide-linked dimeric IRE1$^{Q105C\ SS}$. Shown are data from three independent experiments (mean ± SD). Curve fitting was performed in Prism GraphPad 7.0 using *Equation 2* in Materials and methods (see: *Figure 2—source data 2*).

The online version of this article includes the following source data and figure supplement(s) for figure 2:

**Source data 1.** Source data for *Figure 2A and B*.
**Source data 2.** Source data for *Figure 2D*.
**Figure supplement 1.** Biochemical properties of disulphide-linked dimeric IRE1$^{LD\ Q105C\ SS}$ and monomeric variants used to study MPZ-N's interaction with IRE1$^{LD}$.
**Figure supplement 1—source data 1.** Source data for *Figure 2—figure supplement 1A*.
**Figure supplement 1—source data 2.** Source data for *Figure 2—figure supplement 1D*.
**Figure supplement 2.** Implications of the distance constraint arising from the paramagnetic relaxation enhancement (PRE) experiments with IRE1$^{LD}$ and an MPZ-proxyl-labelled peptide (*Karagöz et al., 2017*) to the possible modes of peptide binding.

exhibited significant protection from exchange (low %ex), whereas surface exposed areas were more dynamic (high %ex) (*Figure 3A* right panel). This method identified the region encompassing residues 303–378 as being especially flexible, a conclusion consistent with the observation that though it was present in the constructs used for crystallisation, residues 308–357 were resolved in neither the crystal structures of wild-type IRE1$^{LD}$ (*Zhou et al., 2006*) (*Figure 3A* right panel dotted line) nor

**Table 1.** Data collection and refinement statistics of IRE1$^{LDQ105C\ SS}$.

**Data collection**

| | |
|---|---|
| **Synchrotron stations** | **Dls i04-1** |
| Space group | P6$_5$22 |
| a,b,c; Å | 182.77, 182.77, 68.45 |
| α, β, γ; $^0$ | 90.00, 90.00, 120.00 |
| Resolution, Å | 91.39–3.55 (3.89–3.55)$^*$ |
| R$_{merge}$ | 0.180 (2.242)$^*$ |
| I/σ(I) | 11.7 (1.5)$^*$ |
| CC1/2 | 1.000 (0.797)$^*$ |
| No. of unique reflections | 8590 (1996)$^*$ |
| Completeness, % | 100.0 (100.0)$^*$ |
| Redundancy | 19.3 (19.9)$^*$ |
| **Refinement** | |
| R$_{work}$/R$_{free}$ | 0.323/0.332 |
| No. of atoms (non H) | 1784 |
| Average B-factors | 127 |
| RMS Bond lengths Å | 0.003 |
| RMS Bond angles,$^0$ | 0.606 |
| Ramachandran favoured region, % | 95.85 |
| Ramachandran outliers, % | 0 |
| MolProbity score† | 1.51 (100$^{th}$) |
| PDB code | 6SHC |

$^*$ Values in parentheses are for highest-resolution shell.
† 100$^{th}$ percentile is the best among structures of comparable resolutions. 0$^{th}$ percentile is the worst.

**Table 2.** List of IRE1[LD] peptic peptides analysed by hydrogen-$^1$H/$^2$H-exchange mass spectrometry (HX-MS) containing the respective m/z values, charge (z) and sequence of each peptide.
Note that the N-terminal amide hydrogen of each peptic fragment exchanges too fast to be detectable with this method. Hence, the N-terminal residue was excluded from the data analysis.

| Residues | M/z | Z | Sequence |
|---|---|---|---|
| 24–36 | 631.345 | 2 | STSTVTLPETLL |
| 37–45 | 938.478 | 1 | FVSTLDGSL |
| 46–59 | 396.730 | 4 | HAVSKRTGSIKWTL |
| 77–85 | 927.435 | 1 | LPDPNDGSL |
| 86–106 | 779.087 | 3 | YTLGSKNNEGLTKLPFTIPEL |
| 96–106 | 636.380 | 2 | LTKLPFTIPEL |
| 107–119 | 1316.680 | 1 | VQASPSRSSDGIL |
| 120–128 | 390.199 | 3 | YMGKKQDIW |
| 130–134 | 735.424 | 1 | YVIDLL |
| 134–145 | 631.832 | 2 | LTGEKQQTLSSA |
| 147–157 | 1090.563 | 1 | ADSLSPSTSLL |
| 157–168 | 730.874 | 2 | LYLGRTEYTITM |
| 168–175 | 522.242 | 2 | MYDTKTRE |
| 176–183 | 535.772 | 2 | LRWNATYF |
| 186–195 | 1031.447 | 1 | AASLPEDDVD |
| 196–208 | 727.837 | 2 | YKMSHFVSNGDGL |
| 209–221 | 703.343 | 2 | VVTVDSESGDVLW |
| 221–232 | 697.856 | 2 | WIQNYASPVVAF |
| 233–240 | 1050.537 | 1 | YVWQREGL |
| 241–248 | 332.864 | 3 | RKVMHINV |
| 253–258 | 406.735 | 2 | LRYLTF |
| 280–287 | 444.28 | 2 | KSKLTPTL |
| 288–296 | 1017.525 | 1 | YVGKYSTSL |
| 297–302 | 655.273 | 1 | YASPSM |
| 303–316 | 474.268 | 3 | VHEGVAVVPRGSTL |
| 317–335 | 956.978 | 2 | PLLEGPQTDGVTIGDKGES |
| 343–360 | 534.307 | 4 | VKFDPGLKSKNKLNYLRN |
| 365–378 | 503.588 | 3 | IGHHETPLSASTKM |
| 379–404 | 516.606 | 6 | LERFPNNLPKHRENVIPADSEKKSFE |
| 410–424 | 810.376 | 2 | VDQTSENAPTTVSRD |
| 410–443 | 727.149 | 5 | VDQTSENAPTTVSRDVEEKPAHAPARPEAPVDSM |

the disulphide-linked IRE1[LD Q105C SS] variant here. Similar characteristics apply to residues 379–444, covering the so-called tail region that connects the structured core of the IRE1[LD] with the transmembrane domain (*Figure 3A* right panel dotted line and *Figure 3—figure supplement 1A*). Moreover, the latter residues overlap with a region of IRE1[LD] implicated in its basal repression in an overexpression cell-based assay (*Oikawa et al., 2007*; *Oikawa et al., 2009*).

To probe the putative loop (residues 308–357) and the tail region (residues 390–444) for their importance in maintaining the repressed state of IRE1 in vivo, we devised a CRISPR-Cas9 mutagenesis strategy (*Figure 3B*). By targeting only unstructured regions within IRE1[LD], we hoped to preserve the integrity of the core structure whilst favouring mutations that might de-repress IRE1 activity. After introducing a set of guide RNAs targeting the region of interest together with the Cas9 endonuclease into cells, error prone non-homologous end joining (NHEJ) resulted in a series of mutations,

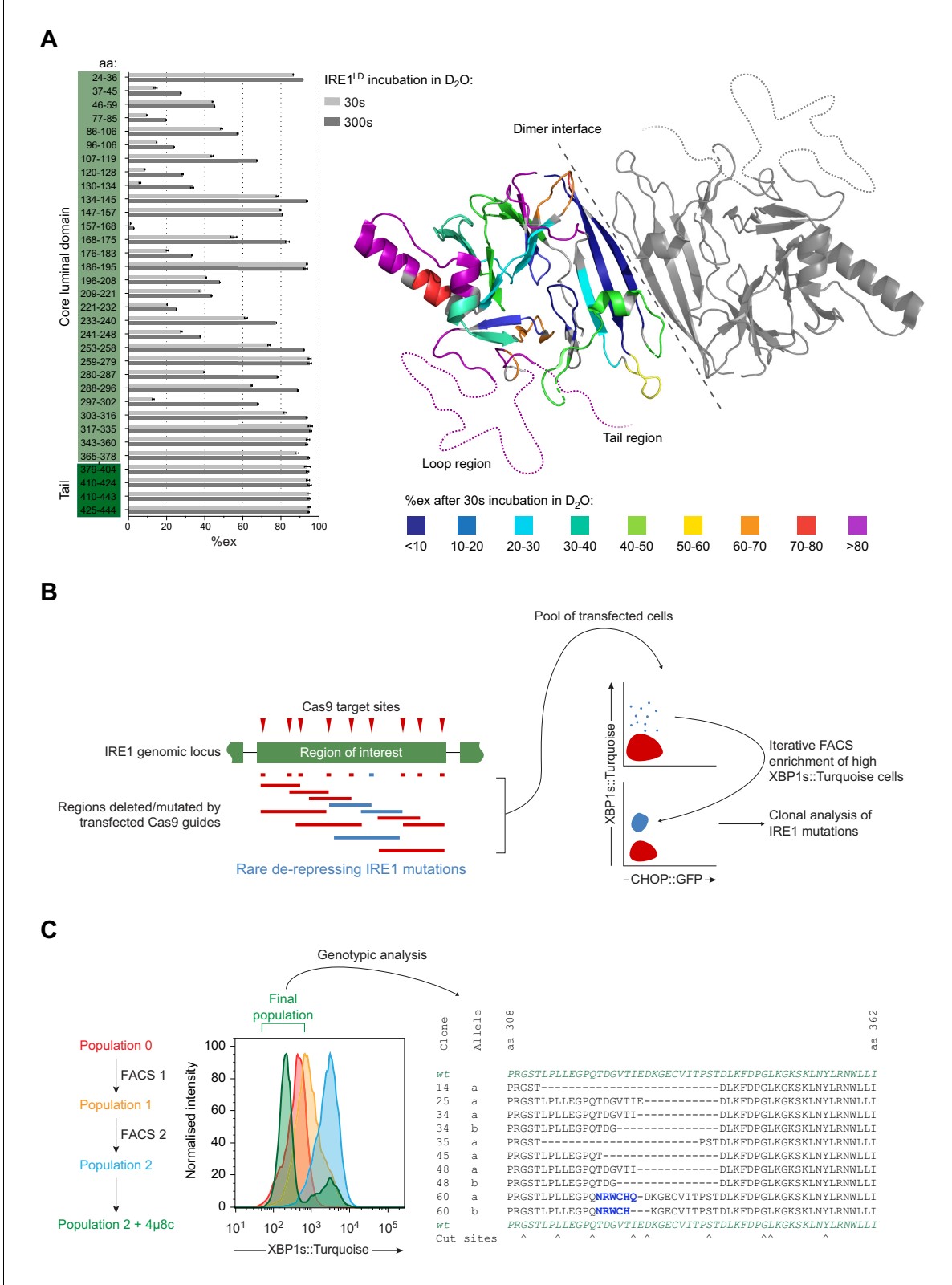

**Figure 3.** Identification of flexible regions in IRE1[LD] that are important for the regulation of IRE1 activity in cells. (**A**) Left panel shows a bar diagram of the percentage of amide hydrogen exchange (%ex) of the indicated by IRE1[LD] segments after 30 and 300 s incubation in $D_2O$. The amino acids (aa) covered by the peptic fragments are indicated on the left. Exchange was corrected for back exchange using a fully deuterated IRE1[LD] preparation. Protein concentration was 5 µM. Shown are the data of three independent experiments (mean ± standard deviation). Right panel shows a cartoon of

*Figure 3 continued on next page*

*Figure 3 continued*

the IRE1$^{LD}$ dimer (PDB: 2HZ6) with the left protomer coloured according to %ex at 30 s (areas with no sequence coverage are uncoloured). The location of the putative loop (residues 308–357) and the tail (residues 390–444) are schematically represented as dotted lines (see: *Figure 3—source data 1*) (B) Schematic description of a directed in vivo CRISPR-Cas9 mutagenesis strategy to probe regions of IRE1$^{LD}$ for their relevance to regulating activity in CHO-K1 cells. Cas9 guides (red triangles) targeted sites across the *Ern1* genomic locus encoding the protein's region of interest. Transfection of individual or pairs of guides resulted in a collection of mutations (insertions and deletions, depicted as blue and red lines). Cell harbouring rare de-repressing mutations of IRE1 (blue) were selected by fluorescence-activated cell sorting (FACS) gated on XBP1s::Turquoise high and CHOP::GFP low signals. The resultant clones were isolated and genotyped. (C) Left panel is a histogram of XBP1s::Turquoise intensity of CHO-K1 dual UPR reporter cell populations transfected with guide-Cas9 encoding plasmids targeting a putative unstructured loop (aa 308–357) within IRE1$^{LD}$ (identified in 'A'). XBP1s:: Turquoise bright cells within population 0 were collected by FACS (FACS1) yielding population 1, followed by a second round of enrichment for bright cells (FACS2 yielding population 2). Population 2 was treated with the IRE1 inhibitor 4μ8c to select against clones exhibiting IRE1-independent reporter activity. The final population was genotypically analysed (representative sequences are shown on the right). Frameshift mutations are coloured in blue and Cas9 cut sites are indicated below.

The online version of this article includes the following source data and figure supplement(s) for figure 3:

**Source data 1.** Source data for *Figure 3A*.
**Figure supplement 1.** IRE1's tail region is involved in maintaining the repressed state of IRE1 in vivo.

ranging from small in-frame indels at a single guide site to larger ones spanning two guide sites. The IRE1 reporter was used to select rare clones exhibiting a de-repressed IRE1 phenotype (XBP1s::Turquoise bright). The CHOP::GFP reporter was used to exclude clones exhibiting a general perturbation of ER protein homeostasis. Iterative rounds of fluorescence-activated cell sorting (FACS) enriched the XBP1s::Turquoise bright population. Clones that had acquired IRE1-independent XBP1s::Turquoise reporter expression were purged based on their unresponsiveness to the IRE1 inhibitor 4μ8c (*Cross et al., 2012*). The *Ern1* locus of individual clones with deregulated IRE1 activity found in the final pool was sequenced (*Figure 3C* left panel). As expected, all the putative deregulating deletions/mutations maintained the frame of the IRE1 coding sequence (*Figure 3C* right panel and *Figure 3—figure supplement 1B*). These observations suggested that deletions of unstructured regions of IRE1$^{LD}$ could deregulate IRE1 activity.

## Characterisation of IRE1$^{LD}$ deletion constructs in vivo and in vitro

To confirm the suggested role of deletions of the loop and the tail region of IRE1$^{LD}$ in deregulating its activity in cells, we reconstituted the endogenous *Ern1* locus of the ΔIRE1 cell line with the most extensive IRE1 deletion variants identified above: the Δloop (missing residues 313–338), the Δtail (missing residues 391–444) or both (ΔΔ). The reconstituted alleles de-repressed IRE1 activity, as indicated by the elevated basal XBP1s::Turquoise signal (*Figure 4A* and *Figure 4—figure supplement 1A*). The IRE1 ΔΔ double deletion had the strongest deregulated phenotype under basal conditions. Like the shorter deletions, the IRE1 ΔΔ double deletion nonetheless retained some responsiveness to stress, albeit with a narrowed dynamic range (*Figure 4A*, compare untreated to tunicamycin-treated samples).

To establish if the deregulating deletion affected the association of the IRE1$^{LD}$ with BiP, we compared the amount of BiP that co-immunoprecipitated with the endogenously expressed wild-type or IRE1 ΔΔ (*Figure 4B*). Despite variation in the total BiP signal intensity in the three independent repeats (*Figure 4B*, lower panel), paired analysis revealed that significantly less BiP was associated with the IRE1 ΔΔ mutant. The same was observed in a transient transfection system in which IRE1's cytosolic effector domains were replaced with glutathione S-transferase (GST). Compared to the wild-type IRE1$^{LD}$-GST bait, the amount of BiP recovered by glutathione affinity chromatography in association with the variants was significantly lower in context of the single deletions and even lower in case of the double-deletion IRE1$^{LD\,ΔΔ}$-GST (*Figure 4C*).

Together, the observations described above confirm a role for the flexible regions of the IRE1$^{LD}$ in maintaining IRE1 in a repressed state in vivo and suggest that such repression may reflect a role for these flexible regions in specifying BiP binding. To follow up on this suggestion, Bio-Layer Interferometry (BLI) was used to compare BiP's association with the biotinylated wild-type or double-deleted IRE1$^{LD\,ΔΔ}$ immobilised on the sensor. Immersing the sensor into a solution containing ERdj4, BiP and ATP gave rise to an association curve, that was reproducibly attenuated when IRE1$^{LD\,ΔΔ}$ was bound as a ligand compared to the wild-type IRE1$^{LD}$ (*Figure 5—figure supplement 1A*, left traces).

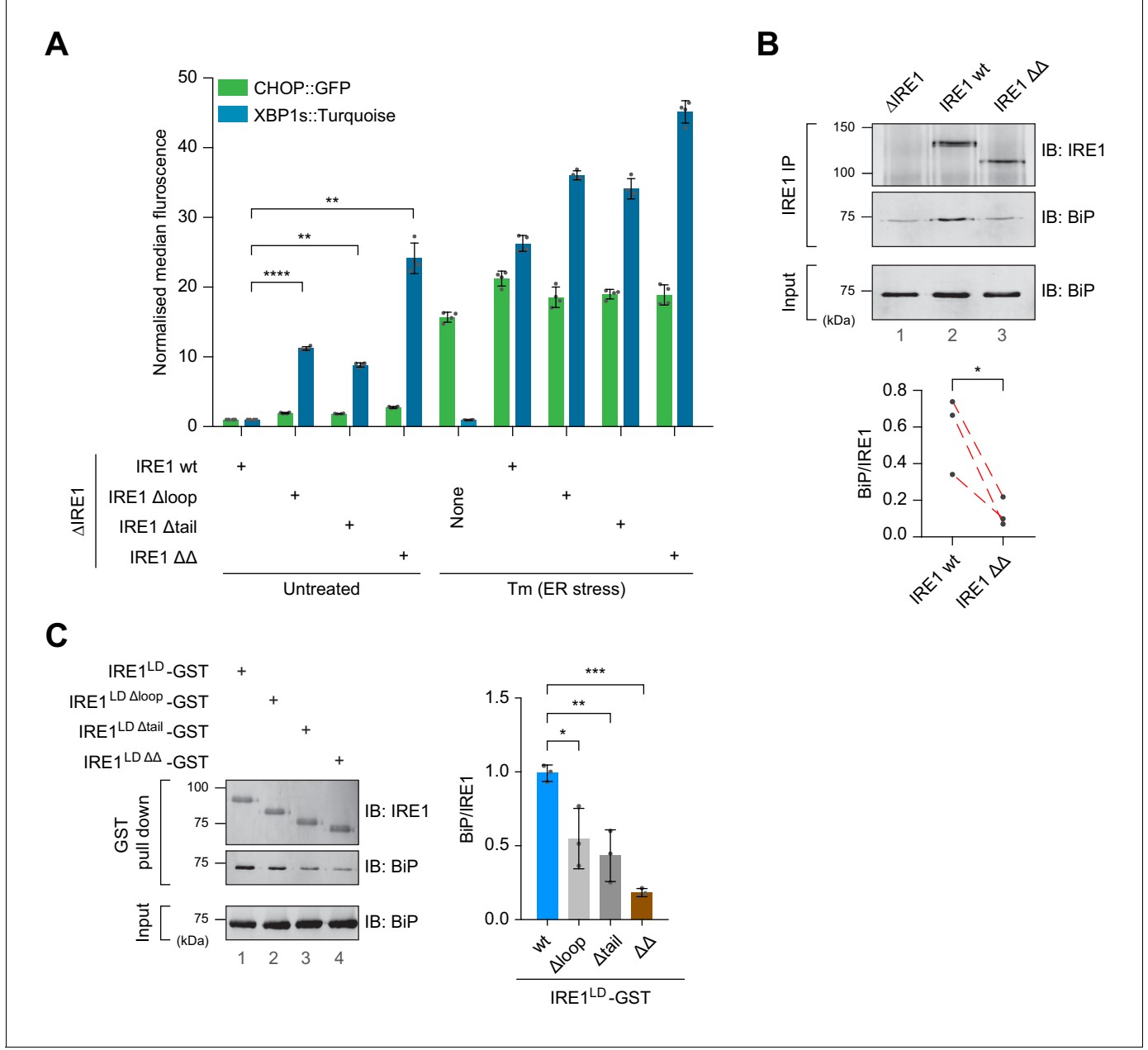

**Figure 4.** Cells expressing IRE1<sup>LD</sup> deletion variants exhibit a de-repressed IRE1 phenotype that correlates with less BiP bound to IRE1. (A) Bar diagram of median XBP1::Turquoise and CHOP::GFP signals from untreated and tunicamycin (Tm)-treated CHO-K1 dual UPR reporter cells with *Ern1* alleles encoding wild-type (wt) or the indicated deletion variants of IRE1 (Δloop, missing residues 313–338, Δtail, missing residues 391–444, or ΔΔ, missing both). Data from four independent experiments is shown [mean ± standard deviation (SD), **: p<0.01, ****: p<0.0001, one-way ANOVA with Sidak's multiple comparison test] (*Figure 4—source data 1*). (B) Representative immunoblot (IB) of endogenously-expressed wt or the IRE1 ΔΔ deletion mutant (see 'A') and associated BiP recovered by immunoprecipitation (IP) of IRE1. BiP in input cell lysates is provided as a loading control. Quantification of the ratio of BiP to IRE1 signals after IP of three independent experiments is shown below (mean ± SD, *: p<0.05, ratio paired parametric Student's t test) (see: *Figure 4—source data 2*) (C) Left panel shows a representative immunoblot the indicated IRE1 variants with glutathione S-transferase (GST) replacing the cytosolic domain. The proteins were introduced into CHO-K1 cells by transient transfection, and the associated endogenous BiP recovered by glutathione pull down. BiP in input cell lysates is provided as a loading control. Quantification of the ratio of BiP to IRE1 signals in the IP of three independent experiments is shown to the right (mean ± SD, *: p<0.05, **: p<0.01, ***: p<0.001, one-way ANOVA with Sidak's multiple comparison test).

The online version of this article includes the following source data and figure supplement(s) for figure 4:

**Source data 1.** Source data for *Figure 4A*.
**Source data 2.** Source data for *Figure 4B*.
*Figure 4 continued on next page*

*Figure 4 continued*

**Figure supplement 1.** Single or double deletion of a flexible loop and the tail within IRE1$^{LD}$ de-repressed IRE1 basal activity.

A similar qualitative defect in BiP binding to IRE1$^{LD\ \Delta\Delta}$ was also observed when the full-length ERdj4 was replaced by its isolated J-domain (that lacks the regions required for specific targeting to the IRE1$^{LD}$) (*Figure 5—figure supplement 1A*, right traces).

This last observation suggested that the defect in J-domain-mediated BiP binding to IRE1$^{LD\ \Delta\Delta}$, had a component that was independent of recruitment of ERdj4 to IRE1$^{LD}$ by the former's targeting domain and implied that the deleted region of IRE1$^{LD}$ had a role in specifying BiP association. This was explored further using J-IRE1$^{LD}$ fusion proteins as BLI ligands to enforce ATP hydrolysis by BiP in proximity to the wild-type or double-deleted IRE1$^{LD}$ (independent of the role these flexible regions of IRE1$^{LD}$ might have in J-domain co-chaperone recruitment). Immersing a BLI sensor loaded either with J-IRE1$^{LD}$ or J-IRE1$^{LD\ \Delta\Delta}$ into a solution containing BiP and ATP revealed a reproducible defect of BiP association to J-IRE1$^{LD\ \Delta\Delta}$ (*Figure 5A* left panel). No association was observed in presence of the substrate binding-deficient BiP$^{V461F}$ mutant. The dissociation in presence of ATP remained similar for both wild-type and J-IRE1$^{LD\ \Delta\Delta}$ (*Figure 5A* right panel), as expected of a process limited by BiP's rate of nucleotide exchange.

The measurements above report on BiP's interaction with the IRE1$^{LD}$ in the context of J-domain-mediated, ATP hydrolysis-driven ultra-affinity (*Misselwitz et al., 1998*; *De Los Rios and Barducci, 2014*). To examine the role of IRE1$^{LD}$'s flexible regions in its affinity for BiP-ADP (an interaction that reports on a segment of the ultra-affinity cycle) we combined BiP with C-terminally biotinylated IRE1$^{LD}$ (either IRE1$^{LD}$-bio or IRE1$^{LD\ \Delta\Delta}$-bio) in presence of ADP and absence of J-domain protein. Given the slow association of BiP-ADP with substrates and the slow dissociation of BiP oligomers a lengthy equilibration (16 hr) was allowed. Almost three-fold less BiP was recovered in complex with IRE1$^{LD\ \Delta\Delta}$-bio than with IRE1$^{LD}$-bio (*Figure 5B*). BiP association was concentration-dependent, destabilised by ATP and was not observed with BiP$^{V461F}$. Coupling of BiP's two domains was dispensable for this interaction with IRE1$^{LD}$-bio, as it was also observed with the domain-uncoupled BiP$^{ADDA}$ (*Preissler et al., 2015a*). Together, these observations point to a role for the flexible regions of IRE1$^{LD}$ in specifying BiP association as a conventional substrate of this Hsp70. An additional role for the flexible regions in ERdj4 recruitment was not evident within the sensitivity of the tools available to us, and therefore remains unexcluded.

BiP binding in vitro promotes dissociation of the IRE1$^{LD}$ dimer (*Amin-Wetzel et al., 2017* and *Figure 1E*). Therefore, we employed the same FRET-based assay to determine if impaired BiP binding affected monomerisation of IRE1$^{LD\ \Delta\Delta}$-containing dimers. Wild-type fluorescent donor-labelled IRE1$^{LD}$ was allowed to dimerise with acceptor-labelled IRE1$^{LD}$ or IRE1$^{LD\ \Delta\Delta}$ and the rate at which BiP, ERdj4 and ATP promoted dissociation of these dimers was measured by following the increase in donor fluorescence over time. The initial velocity of BiP-mediated monomerisation of the IRE1$^{LD\ \Delta\Delta}$ containing heterodimers was considerably slower than monomerisation of wild-type homodimers (*Figure 5C*).

In BLI experiments, BiP association to monomeric IRE1$^{LD\ P108A}$ was faster than to the enforced dimeric IRE1$^{LD\ Q105C\ SS}$ (*Figure 5—figure supplement 1B*), raising the concern that both diminished BiP binding to the IRE1$^{LD\ \Delta\Delta}$ observed in BLI (*Figure 5A*) and the slower monomerisation of the IRE1$^{LD\ \Delta\Delta}$ containing FRET pair (*Figure 5C*) might reflect intrinsically enhanced stability of the IRE1$^{LD\ \Delta\Delta}$-containing dimers. However, SEC of the purified proteins performed over a range of protein concentrations reported on similar affinities of the wild-type and IRE1$^{LD\ \Delta\Delta}$ dimers (*Figure 5—figure supplement 1C and D*) yielding K$_{1/2\ max}$ values in the same order of magnitude as the K$_D$ of dimerisation measured by AUC (*Zhou et al., 2006*). Together, these observations suggest that diminished BiP binding to IRE1$^{LD\ \Delta\Delta}$ resulted in an impairment of BiP-driven IRE1$^{LD}$ monomerisation.

## BiP-driven monomerisation of IRE1$^{LD}$ assessed by HX-MS

To complement the kinetic observations pointing to impaired BiP-driven monomerisation of IRE1$^{LD\ \Delta\Delta}$ with structural correlations, HX-MS was performed. To establish the HX-MS signature of monomerisation, deuteron incorporation was compared between wild-type and dimerisation-defective IRE1$^{LD\ W125A}$ or IRE1$^{LD\ P108A}$ mutants. This reported on monomerisation-induced deprotection

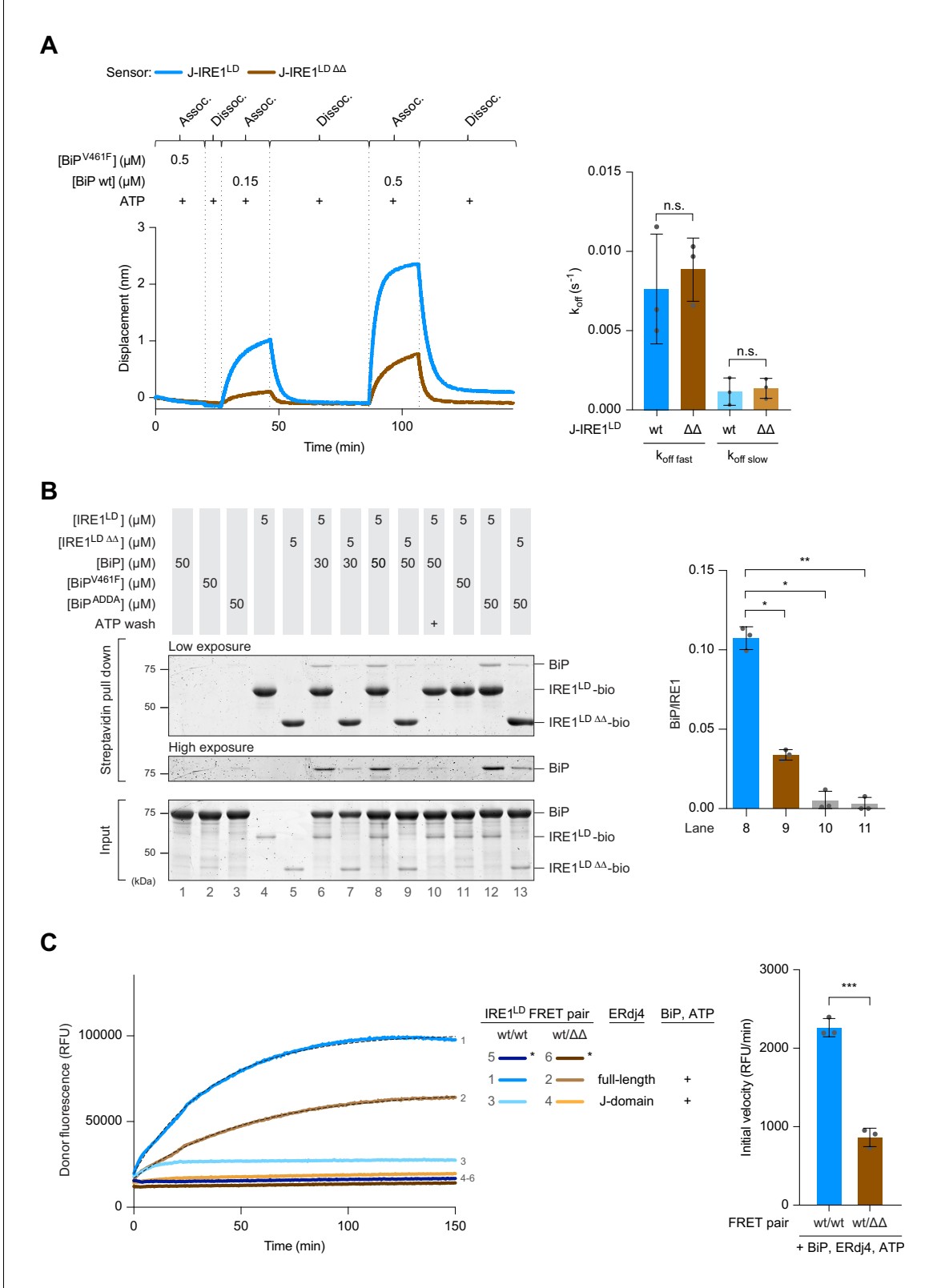

**Figure 5.** Impaired BiP binding and monomerisation of IRE1$^{LD\,\Delta\Delta}$ in vitro. (**A**) Left panel shows Bio-Layer Interferometry (BLI)-derived association (assoc.) and dissociation (dissoc.) traces of streptavidin sensors loaded with the indicated biotinylated ligands [a fusion of ERdj4's J-domain to IRE1$^{LD}$ wild-type (wt) or ΔΔ, as in *Figure 1D*] and exposed sequentially to the indicated solutions of analyte (containing wt BiP or the client-binding mutant BiP$^{V461F}$). A representative experiment of three independent repetitions is shown. The traces were subtracted against a background derived from a BLI sensor with

*Figure 5 continued on next page*

*Figure 5 continued*

no ligand and the BLI signals (displacement) were set to zero after the first washing step. Quantification of the dissociation rate constants $k_{off}$ after association in presence of 0.15 µM BiP and 2 mM ATP are shown to the right. Traces were fitted to a two-phase dissociation function in Prism GraphPad 7.0. Shown are the mean ± standard deviation (SD) of three independent repetitions (n.s.: not significant, unpaired parametric Student's t test). **(B)** Coomassie-stained SDS-PAGE gel of biotinylated wt IRE1$^{LD}$ (IRE1$^{LD}$-bio), double deleted IRE1$^{LD\ \Delta\Delta}$ (IRE1$^{LD\ \Delta\Delta}$-bio) and BiP, recovered on a streptavidin matrix from samples constituted as indicated. 2 mM ATP was used during wash steps of the matrix when indicated. A representative data set is shown. Quantification of the ratio of BiP to IRE1 signals in the relevant samples after pull down from three independent experiments is shown on the right (mean ± SD, *: p<0.05, **: p<0.01, one-way ANOVA with Sidak's multiple comparison test). **(C)** Time-dependent change in donor fluorescence of the indicated IRE1$^{LD}$ FRET pair incubated at t = 0 with the components shown to the right. The asterisks mark samples set up with a mock FRET sensor lacking the IRE1$^{LD}$ donor-labelled molecule. Protein concentrations were 0.2 µM FRET pair, 30 µM BiP, 2.5 µM full-length ERdj4 (or its isolated J-domain) and 2 mM ATP. A representative experiment of three independent repetitions is shown. When indicated, the data points were fitted to a one-phase association function in Prism GraphPad 7.0; the initial velocity represents the slope of the curve at time point zero (mean ± SD, ***: p<0.001, unpaired parametric Student's t test) (see: *Figure 5—source data 1*).

The online version of this article includes the following source data and figure supplement(s) for figure 5:

**Source data 1.** Source data for *Figure 5A.*
**Source data 2.** Source data for *Figure 5C.*
**Figure supplement 1.** The ΔΔ deletion does not affect the stability of the IRE1$^{LD}$ dimer.
**Figure supplement 1—source data 1.** Source data for *Figure 5—figure supplement 1B.*
**Figure supplement 1—source data 2.** Source data for *Figure 5—figure supplement 1C.*

of several peptic peptides from the IRE1$^{LD}$ (*Figure 6A*). Projecting these areas onto the crystal structure revealed that monomerisation affected HX at the dimer interface but also in parts further away (*Figure 6B*). IRE1$^{LD}$ monomerisation thus induced structural rearrangements across the protein. Moreover, the difference plot in HX reported on a gradation between both mutant variants, as IRE1$^{LD\ P108A}$ had an enhanced signature of monomerisation compared to IRE1$^{LD\ W125A}$, matching the hierarchy of dimer instability observed by SEC and DSF analysis (*Figure 2—figure supplement 1A and D*, respectively).

A similar deprotection signature, affecting most of the peptic peptides that are exposed upon monomerisation, was observed when IRE1$^{LD}$ was incubated with BiP and ERdj4 in presence of ATP (*Figure 6C* upper row, box 1: residues 77–128 and box 2: residues 280–302). Monomerisation was dependent on the integrity of all components of the reaction, as neither the substrate binding BiP$^{V461F}$ mutant nor the ERdj4$^{QPD}$ supported the pattern of deprotection observed in the monomeric versions of IRE1$^{LD}$ (the significance of the protection afforded by BiP$^{V461F}$ and ERdj4$^{QPD}$ to some peptides is presently unknown). IRE1$^{LD\ \Delta\Delta}$ exhibited delayed monomerisation in presence of BiP, ERdj4 and ATP: IRE1$^{LD\ \Delta\Delta}$'s signature of monomerisation was absent after 30 s incubation in D$_2$O (*Figure 6C* lower row) and was faint even after an exchange reaction of 300 s (*Figure 6—figure supplement 1B and C* lower row). In the absence of BiP, ERdj4 and ATP the difference plot comparing deuteron incorporation into IRE1$^{LD}$ and IRE1$^{LD\ \Delta\Delta}$ was negligible (*Figure 6—figure supplement 1D*) providing independent confirmation of the SEC measurements pointing to similar stability of the wild-type and IRE1$^{LD\ \Delta\Delta}$ mutant dimer (*Figure 5—figure supplement 1C and D*). Thus, HX-MS provided an orthogonal assay to the FRET-based measurement, reporting on BiP-mediated monomerisation of IRE1$^{LD}$ and a kinetic defect in this process brought about by deletion of flexible regions in the luminal domain that enforce IRE1's repressed state in cells.

Close inspection of the HX-MS data revealed that some of the peptides (e.g. peptides 636.380$^{2+}$ and 655.273$^{+}$ corresponding to residues 96–106 and 297–302, respectively) exhibited clear bimodal isotope distribution. This characteristic is a signature for the EX1 exchange regime, indicative of the presence of two discrete subpopulations of molecules: a more folded and therefore low exchanging subpopulation and a more open, high exchanging subpopulation (*Figure 7—figure supplement 1A* and Materials and methods section). The contribution of low and high exchanging subpopulations to each isotope peak was determined by fitting the isotope peak maxima versus m/z data points (*Figure 7A*) to a two Gaussian distribution model (*Hentze et al., 2016*). From the fit parameters the fraction of each isotope peak that belongs to the low and high exchanging subpopulation was calculated [*Figure 7—figure supplement 1A* (blue and red parts of the bars) and 1B]. Comparison with the unexchanged and the 100% control samples revealed that the low exchanging subpopulation was largely protected from HX, whereas the high exchanging subpopulation had almost all amide

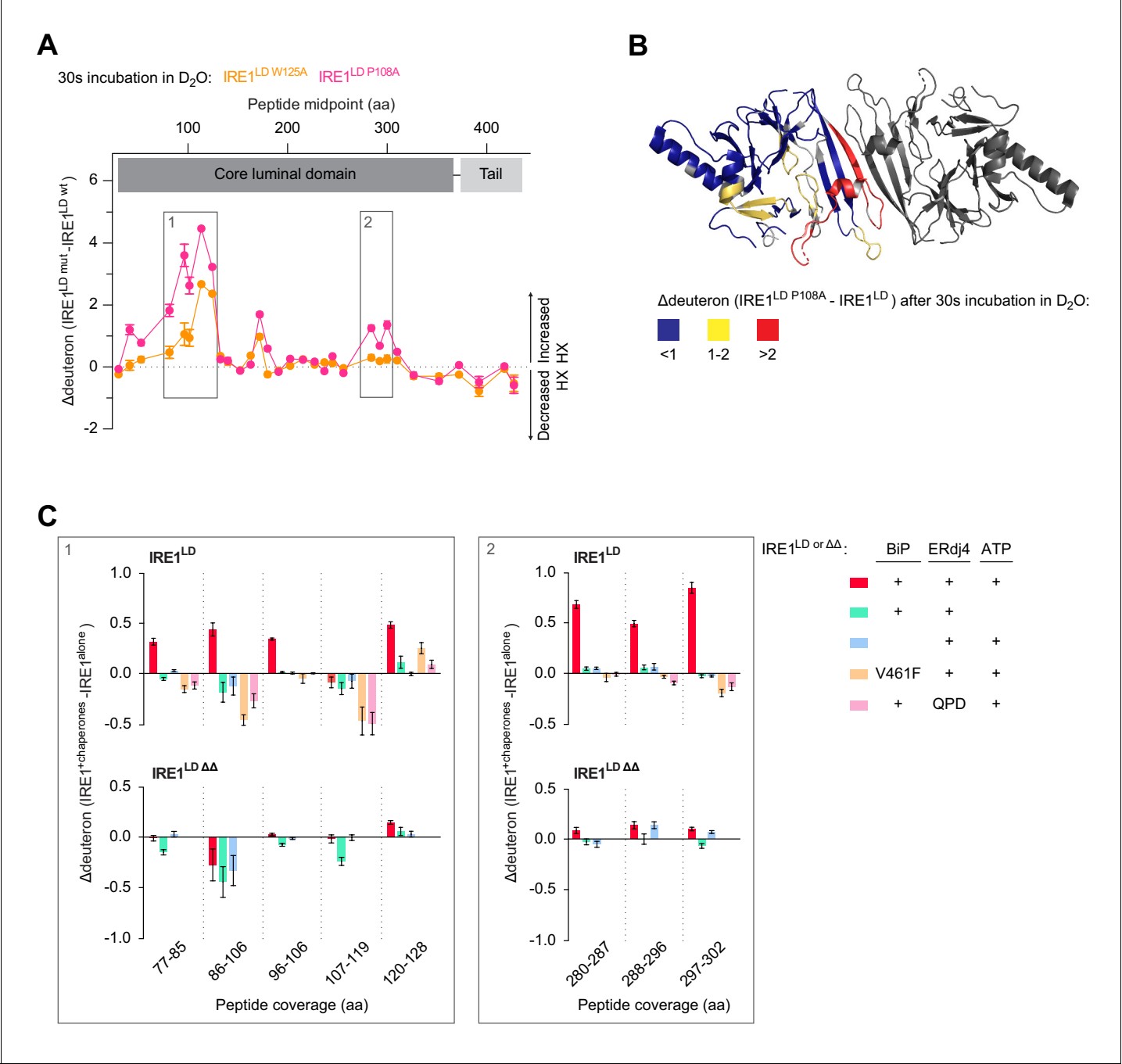

**Figure 6.** BiP-mediated monomerisation of IRE1^LD ΔΔ assessed by hydrogen exchange mass spectrometry (HX-MS). (**A**) Difference plot of deuteron incorporation comparing wild-type (wt) IRE1^LD with the monomeric mutants IRE1^LD W125A (orange trace) or IRE1^LD P108A (pink trace) after 30 s incubation in D₂O [see *Table 2* for the amino acid (aa) sequence of the individual segments]. Protein concentration was 5 μM. Shown are data from three independent experiments [mean ± standard deviation (SD)]. Boxes 1 and 2 highlight regions of greater hydrogen exchange (HX) in the monomeric mutants compared to wt IRE1^LD that were analysed in presence of chaperones in 'C' (see *Figure 6—source data 1*). (**B**) Cartoon representation of the IRE1^LD dimer (PDB: 2HZ6) coloured according to the difference of deuteron incorporation between wt and IRE1^LD P108A after 30 s of incubation in D₂O (from 'A'). (**C**) Difference plot of the deuteron incorporation between the untreated sample and samples exposed to the indicated additives. The data for the same peptic peptides from wt IRE1^LD and the IRE1^LD ΔΔ mutant are displayed separately. Protein concentrations were 5 μM IRE1^LD (wt or ΔΔ mutant), 30 μM BiP (wt or V461F mutant), 6 μM ERdj4 (wt or QPD mutant) and 2 mM ATP. Shown are the means ± SD of three data sets acquired after 30 s incubation in D₂O (the corresponding 300 s data set is presented in *Figure 6—figure supplement 1A and C*) (see *Figure 6—source data 2*). The online version of this article includes the following source data and figure supplement(s) for figure 6:

**Source data 1.** Source data for *Figure 6A* and *Figure 6—figure supplement 1A*.

*Figure 6 continued on next page*

*Figure 6 continued*

**Source data 2.** Source data for *Figure 6C* and *Figure 6—figure supplement 1C.*
**Figure supplement 1.** HX-MS evidence for impaired BiP- and ERdj4-driven monomerisation of IRE1$^{LD\ \Delta\Delta}$.
**Figure supplement 1—source data 1.** Source data for *Figure 6—figure supplement 1E.*

protons exchanged for deuterons. Moreover, the low exchanging subpopulation converted into the high exchanging subpopulation with time (*Figure 7—figure supplement 1A*, compare 30 and 300 s incubation in $D_2O$).

Interestingly, the degree of conversion into the high exchanging subpopulation was more pronounced for the IRE1$^{LD\ P108A}$ monomeric mutant than for wild-type IRE1$^{LD}$ and essentially complete after 300 s (*Figure 7—figure supplement 1A* left panel). SEC analysis of IRE1$^{LD\ P108A}$ showed that at 5 µM (the concentration at which the protein was diluted into $D_2O$) it is mostly monomeric (*Figure 2—figure supplement 1A*). Hence, these data suggest that the conversion from the low exchanging subpopulation to the high exchanging subpopulation was a feature of the monomeric state.

HX is a quasi-irreversible reaction: Once a molecule has transiently assumed a high exchanging conformation (and undergone the exchange) the signature of having transited through a high exchanging conformation remains even if the protein is in a conformational equilibrium (and individual molecules transit back to the low exchanging conformation). Thus, HX-MS detects the transition to the high exchange endpoint. The observation that for wild-type IRE1$^{LD}$ the transition from the low (blue) to the high (red) exchanging population occurred with much slower kinetics than for IRE1$^{LD\ P108A}$ (*Figure 7—figure supplement 1A*, compare left panel, monomeric IRE1$^{LD\ P108A}$ with the right panel, wild-type IRE1$^{LD}$) suggests that a higher proportion of IRE1$^{LD}$ monomers increased the transition rate, whereas the presence of IRE1$^{LD}$ dimers leads to a reduction of the rate constant. Hence, the extracted transition rate $k_{trans}$ reports on IRE1$^{LD}$'s monomer-dimer equilibrium during the reaction.

Next, we compared the $k_{trans}$ of peptic peptide 655.273$^+$ from wild-type IRE1$^{LD}$ in presence and absence of BiP, ERdj4 and ATP. Due to pre-incubation of the reactions, the three-protein system already had a higher proportion of monomeric IRE1$^{LD}$ at the point of dilution into $D_2O$ (reflected in a greater proportion of the high mass population at the earliest measurement). Nevertheless, an accelerated time-dependent increase in the proportion of monomeric IRE1$^{LD}$ was observed in the BiP-treated sample, indicating an increase in $k_{trans}$ (*Figure 7A and B*). Acceleration of $k_{trans}$ was also observed with peptide 636.380$^{2+}$ in presence of BiP, ERdj4 and ATP (*Figure 7C* and *Figure 7—figure supplement 1C*).

Because it is affected by peptide-specific flexibility, $k_{trans}$ itself is not a direct measure of the first order dissociation rate of the IRE1$^{LD}$ dimer (its $k_{off}$), however, the difference observed in $k_{trans}$ for any individual peptide measured under two conditions mainly reports on differences in IRE1$^{LD}$ dimer dissociation. Therefore, these findings imply that BiP-induced IRE1$^{LD}$ monomerisation has a component arising from active destabilisation of the dimer.

## Discussion

The notion that a chaperone machinery with an Hsp70, such as BiP, as its terminal effector might negatively regulate activity of an upstream UPR transducer, such as IRE1, has the appeal of simplicity: Hsp70's can potently affect the structure and function of their clients. The level of free BiP is kept low by inactivating oligomerisation and AMPylation and is further limited by client titration (*Preissler and Ron, 2019*). Therefore, the availability of a BiP-dependent machinery to serve as an active repressor of IRE1 is a plausible inverse measure of the level of ER stress. For years, the inverse relationship between the recovery of BiP in complex with IRE1 and exposure of cells to conditions causing ER stress has provided the only experimental support for this chaperone repression model (*Bertolotti et al., 2000*; *Okamura et al., 2000*; *Oikawa et al., 2009*). The recent establishment of an ATP- and co-chaperone-dependent system in which BiP promotes a pool of monomeric, inactive-state IRE1$^{LD}$ further supports the model by revealing BiP's potential to affect a major change in IRE1's activity in vitro (*Amin-Wetzel et al., 2017*). Here, we provide much needed further support

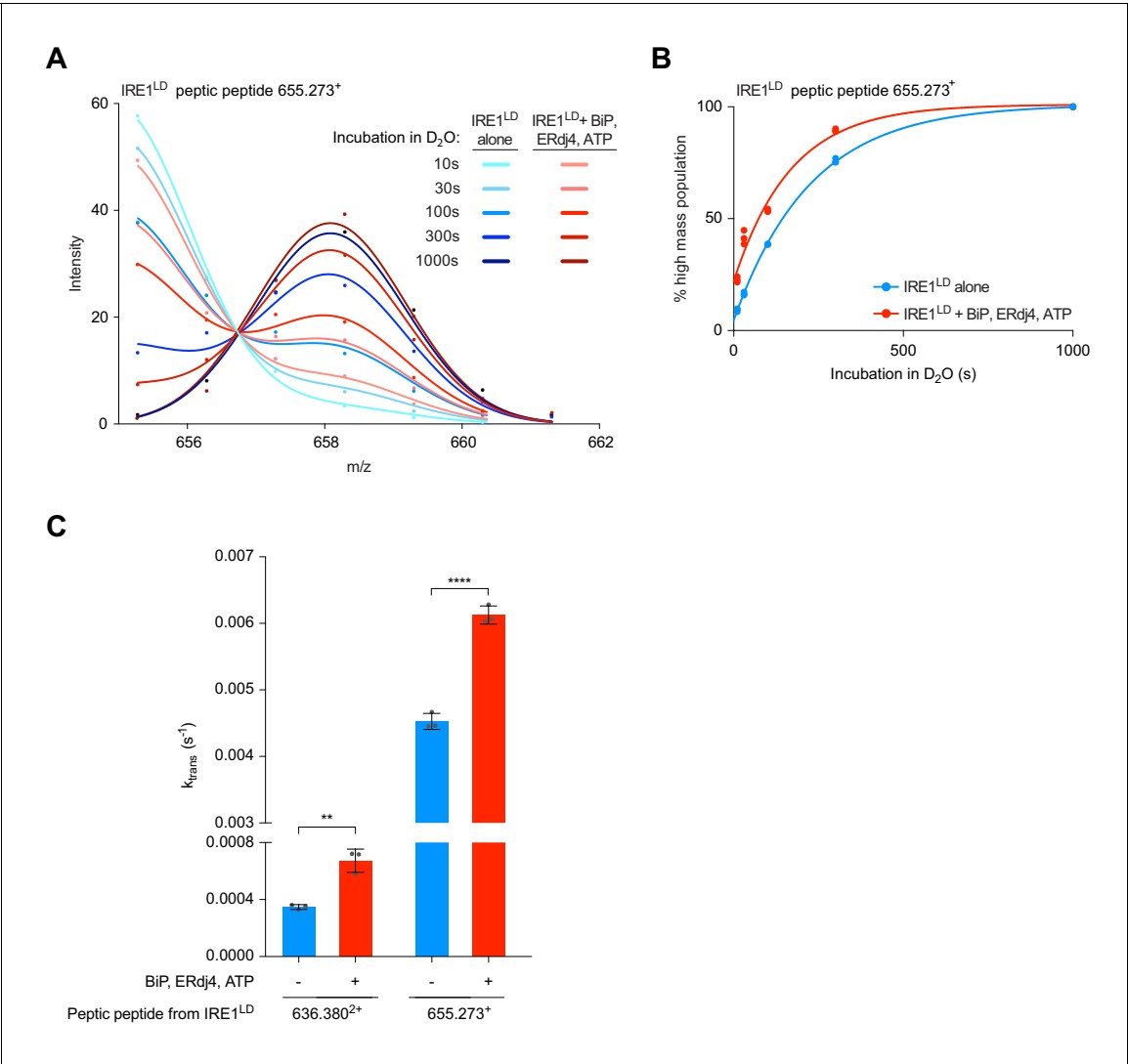

**Figure 7.** Analysis of bimodally-distributed isotope clusters of IRE1$^{LD}$ peptic peptides reveals active destabilisation of the IRE1$^{LD}$ dimer by BiP. (**A**) Intensity distributions of the isotope clusters of peptide 655.273$^{+}$ (residues 297–302) from IRE1$^{LD}$, untreated or exposed to BiP, ERDj4 and ATP (30 min at 30˚C) following different incubation times in D$_2$O, as indicated. Curves are fits of the sum of two Gaussian distributions (Prism GraphPad 7.0, see *Equation 4* in Materials and methods). A representative plot of three independent experiments is shown. (see: *Figure 7—source data 1*) (**B**) Plot of time-dependent change in the fractional contribution of high mass species to the isotope clusters of peptide 655.273$^{+}$ (from 'A') calculated as described in *Figure 7—figure supplement 1A and B*. Shown are data points from three independent samples of IRE1$^{LD}$ in presence and absence of BiP, ERdj4 and ATP. The curves were fitted to a one-phase association model in Prism GraphPad 7.0. Data for a second informative peptide is shown in *Figure 7—figure supplement 1C*. (**C**) Bar diagram of the transition rate constant k$_{trans}$ extracted by analysis of bimodal distributions in the isotope clusters of peptic fragments 636.380$^{2+}$ and 655.273$^{+}$ from IRE1$^{LD}$ in presence and absence of BiP, ERdj4 and ATP (from *Figure 7B* and *Figure 7—figure supplement 1C*). All the data points from three independent experiments are shown and the mean ± standard deviation (\*\*: p<0.01, \*\*\*\*: p<0.0001, one-way ANOVA with Sidak's multiple comparison test).

The online version of this article includes the following source data and figure supplement(s) for figure 7:

**Source data 1.** Source data for *Figure 7A and B* and *Figure 7—figure supplement 1A, B, C*.

**Figure supplement 1.** Analysis of the bimodal distributions of the isotope clusters detected by HX-MS.

for the chaperone repression model by demonstrating that directing endogenous BiP to bind endogenous IRE1$^{LD}$ as a substrate also attenuates signalling in cells, thus revealing BiP's potential as a direct IRE1 repressor in vivo.

A structure-based targeted approach identified regions of IRE1$^{LD}$ that impart a repressed state in vivo. The same regions proved important for ATP and co-chaperone-dependent BiP-mediated

conversion of active-state IRE1$^{LD}$ dimers to inactive-state monomers in vitro and their presence accelerated the formation of an ATP and co-chaperone-dependent complex with BiP in vitro. Monomerisation was observed in both a FRET-based assay, involving labelled molecules of IRE1$^{LD}$, and in an HX-MS assay with intact molecules, thus establishing a firm correlation between the determinants of IRE1 that regulate its function in vivo and those that specify its regulation in vitro by a BiP-led machinery (*Figure 8*).

Correlation between factors involved in BiP regulation of IRE1 in vitro and UPR activity in vivo have been previously noted: Deregulated AMPylation of BiP activates IRE1 in cells (*Preissler et al., 2015b*) and BiP AMPylation in vitro blocks IRE1$^{LD}$ monomerisation (*Amin-Wetzel et al., 2017*). ERdj4 acts in concert with BiP to monomerise IRE1$^{LD}$in vitro and loss of ERdj4 from cells derepresses IRE1 in vivo (*Amin-Wetzel et al., 2017*). However, genetic lesions in trans-acting ER-localised factors also have the potential to broadly alter the state of the ER and thereby unleash processes that affect IRE1 independently (of any direct interaction with BiP). Indirect effects are less likely a consequence when IRE1$^{LD}$ is modified in cis. Therefore, whilst it is impossible to rule out contributions from factors other than the BiP machinery to the deregulation of IRE1 that arises from deletion of the unstructured regions of its luminal domain, attenuation of BiP-mediated IRE1 repression in cells emerges as a parsimonious unifying explanation for the findings presented here.

It is further notable that there is nothing in our observations to speak against the possibility that extended regions of unfolded ER proteins serve as activating ligands of IRE1 by binding across the IRE1$^{LD}$ dimer interface and stabilising it (*Karagöz et al., 2019*). IRE1 signalling is triggered by an imbalance between unfolded proteins and BiP. The latter results in more potential ligands for IRE1 and fewer molecules of its client-free ATP bound BiP repressor (*Bakunts et al., 2017*; *Vitale et al., 2019*). Thus, the two proposed mechanisms for IRE1 activation, could well co-exist. However, our findings do raise questions regarding the strength of the experimental evidence supporting the current ideas how unfolded proteins may serve as activating ligands of IRE1. The evidence rests prominently on the activity of a peptide, MPZ-N, nominated as a model activating ligand of IRE1$^{LD}$ (*Karagöz et al., 2017*). Our findings do not support the notion that this peptide specifically engages

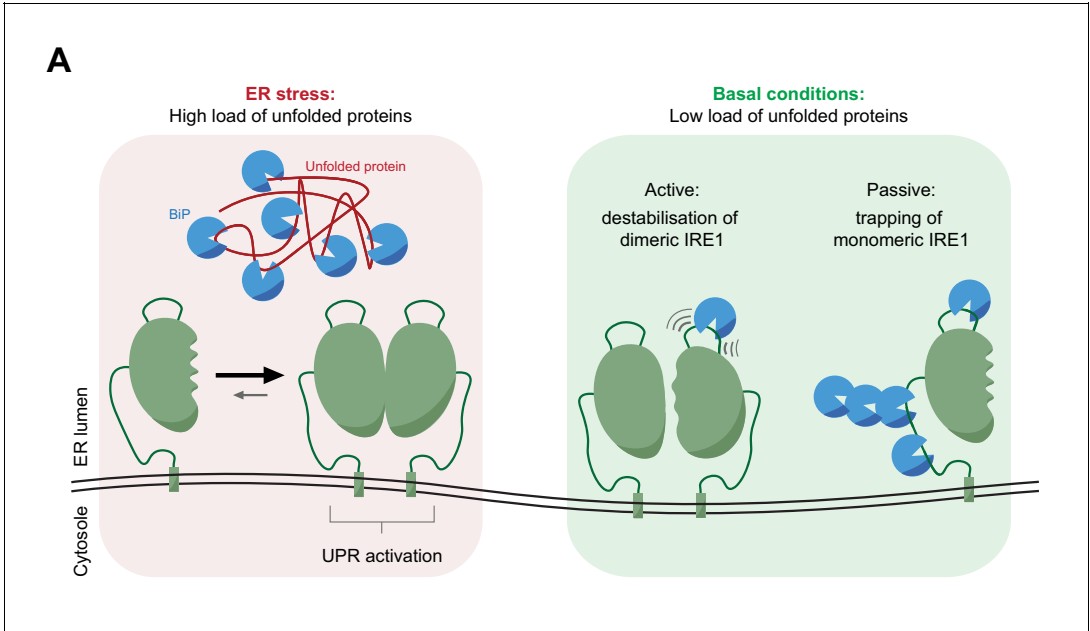

**Figure 8.** Cartoon depicting features of BiP-mediated regulation of IRE1 activity. In stressed cells unfolded proteins compete for BiP, exposing IRE1$^{LD}$ to a default dimeric active state, specified by the kinetics of the monomer-dimer equilibrium (left panel). In compensated cells BiP (assisted by ERdj4 and possibly other J-domain proteins, not shown) binds flexible regions of IRE1$^{LD}$. Engagement of these regions in the IRE1$^{LD}$ dimer may favor active dimer disassembly by entropic pulling or allosterically induced conformational changes (right panel). BiP binding to the same flexible regions of the IRE1$^{LD}$ monomer, may inactivate IRE1 by disfavoring re-dimerisation (right panel). The dynamic nature of BiP binding, which entails cycles for ATP hydrolysis-driven client engagement and nucleotide exchange-mediated release, ensures that IRE1 activity is kinetically coupled to the balance between unfolded protein load and folding capacity of the cell.

the MHC-like groove traversing the dimer interface, as a disulphide, crystallographically proven to lie across this groove, thereby locking the helices in a 'closed' conformation, had no effect on the binding of MPZ-N to IRE1$^{LD}$. Furthermore, MPZ-N binding to IRE1$^{LD}$ did not stabilise it thermodynamically, whereas the aforementioned disulphide, which mimics the proposed dimer-stabilising effect of a bound peptide, increased the melting temperature IRE1$^{LD}$ by 10 °C. Nor did MPZ-N promote a shift in the monomer-dimer equilibrium of IRE1$^{LD}$ as assessed by SEC. These concerns, along with the lack of crystallographic data supporting engagement of the groove by ligands, suggest the need for further experiments to test the role of unfolded proteins as direct IRE1 activators.

HX-MS analysis revealed neither an ERdj4-dependent nor BiP and ATP-dependent protection within IRE1$^{LD}$ to suggest their binding site. It has been proposed that ERdj4's bacterial homolog, DnaJ, exploits mostly side chain interactions to bind clients, with a strong preference for aromatic residues (*Rüdiger et al., 2001*). Such interactions, were they to serve as the basis for IRE1$^{LD}$ recognition by ERdj4, would be only visible to HX-MS if they stabilised the underlying secondary structure. Given the conventional mode of BiP action on IRE1$^{LD}$ (ATP and co-chaperone-dependent and abolished by the BiP$^{V461F}$ mutation), one would expect protection of 4–5 hydrogen amides by IRE1$^{LD}$ engagement in the chaperone's substrate-binding domain. However, partial occupancy of multiple sites may have diluted any HX-MS signature of BiP binding. This is supported by the observation that at the concentrations of ERdj4 and BiP used in the HX-MS assay, the FRET assay reported peak fluorescence of only ~40% of the unquenched donor (at the kinetically driven pseudo steady state plateau of the reaction, *Figure 1E*). Thus, the lack of a clear ATP- and ERdj4-dependent BiP binding signature is consistent with the dynamic nature of BiP's interaction with IRE1$^{LD}$. Interestingly, we observed an ATP-independent protection against deuteron incorporation within IRE1$^{LD}$ that was also evident in presence of the substrate binding-defective BiP$^{V461F}$. This might reflect a non-conventional interaction of BiP's nucleotide binding domain with IRE1$^{LD}$, as proposed by the Ali lab (*Kopp et al., 2018*; *Kopp et al., 2019*). However, this protection is not correlated to the activity-state of IRE1$^{LD}$ and its significance thus remains to be established.

Mechanistically, BiP's interaction with IRE1$^{LD}$ shares features with other situations in which Hsp70s bind to native clients thereby regulating their activity: DnaJ-directed, DnaK-mediated destabilisation of *E. coli* σ$^{32}$ (*Rodriguez et al., 2008*), functional regulation of the glucocorticoid receptor (*Kirschke et al., 2014*), regulation of the activity of the tumour suppressor p53 (*Boysen et al., 2019*; *Dahiya et al., 2019*), Hsf1 regulated heat shock gene expression (*Abravaya et al., 1992*) and Hsc70-mediated destabilisation of clathrin coats (*Sousa et al., 2016*). All these have in common client destabilisation and likely initiate at unstructured regions of the substrate. Thus, it seems reasonable to suggest that an important aspect of BiP's ability to affect the disposition of IRE1$^{LD}$'s monomer-dimer equilibrium arises from its interaction with the flexible regions identified here.

Bimodal analysis of the HX-MS data suggested that ERdj4-directed BiP binding can accelerate dimer disassembly. This is consistent with the ability of the IRE1$^{LD}$ dimer to serve as a ligand for ERdj4 and BiP (here and *Amin-Wetzel et al., 2017*). While BiP binding to and stabilisation of IRE1$^{LD}$ monomers may also contribute to shifting the monomer-dimer equilibrium towards the former, the HX-MS experiment suggests an (additional) active role for BiP in dimer destabilisation. This may arise from a BiP-binding induced bias of the ensemble of IRE1$^{LD}$ dimers towards conformers preferentially populated in the monomer. A similar mechanism of conformational selection has been proposed for DnaK-mediated destabilisation of *E. coli* σ$^{32}$ (*Rodriguez et al., 2008*). Such 'allosteric' action is consistent with the observation that monomerisation has effects on IRE1$^{LD}$ structure that are far removed from the dimer interface. Alternatively, BiP binding may destabilise the IRE1$^{LD}$ dimer by entropic pulling (*De Los Rios et al., 2006*), as has been suggested in Hsc70-mediated destabilisation of clathrin coats (*Sousa et al., 2016*) (*Figure 8*). The latter mechanism would be further favoured by assembly of BiP oligomers on the surface of the IRE1$^{LD}$, a possibility consistent with the >1:1 stoichiometry of BiP:IRE1$^{LD}$ complexes observed in some experiments (*Amin-Wetzel et al., 2017*) (although the latter may also reflect multiple BiP binding sites).

As shown here, flexible regions of IRE1$^{LD}$ contribute measurably to its repression in cells and to BiP-driven monomerisation in vitro. This observation is consistent with the idea that these regions serve as initiation points for BiP binding to promote dimer disassembly via entropic pulling, allosterically induced conformational changes or both. Considerable redundancy seems built into the process, as the deregulated IRE1$^{\Delta\Delta}$ allele retained a measure of stress responsiveness in cells and the IRE1$^{LD}$ $^{\Delta\Delta}$ dimer was still slowly undone in a BiP-dependent process in vitro. Such redundancy has

been observed previously: in both yeast and human, IRE1 deletion of the tail region connecting the structured core of IRE1$^{LD}$ to the transmembrane domain partially deregulated IRE1, whilst retaining partial responsiveness to ER stress (*Oikawa et al., 2007*; *Oikawa et al., 2009*). Redundancy in the structural features of the IRE1$^{LD}$ dimer that render it a substrate for BiP-dependent disassembly and the non-equilibrium kinetic nature for BiP's action could serve as the basis for a smoothly graded response to variation in the levels of ER stress.

# Materials and methods

## Key resources table

| Reagent type (species) or resource | Designation | Source or reference | Identifiers | Additional information |
|---|---|---|---|---|
| Strain, strain background (*Escherichia coli*) | BL21 C3013 *E. coli* | NEB | Cat no: C3013I | |
| Strain, strain background (*Escherichia coli*) | Origami B(DE3) *E. coli* | Novagen/MERCK | Cat no: 70837 | |
| Antibody | Anti-mouse IRE1α serum (rabbit polyclonal) | *Bertolotti et al., 2000* | NY200 | used at 1/1000 |
| Antibody | anti-hamster BiP (chicken polyclonal) | *Avezov et al., 2013* | Anti-BiP | used at 1/1000 |
| Antibody | Anti-GST (polyclonal rabbit) | *Ron and Habener, 1992* | Anti-CHOP | used at 1/1000 |
| Cell line, (*Cricetulus griseus*) | Clone S21 a derivative of RRID: CVCL_0214 | *Sekine et al., 2016* | CHO-K1 S21 | CHO CHOP::GFP, XBP1s::Turquoise dual UPR reporter cell line |
| Cell line, (*Cricetulus griseus*) | CHO-K1 S21 CHOP::GFP, XBP1s::Turquoise ΔLD 15 | *Kono et al., 2017* | ΔIRE1 | CHO CHOP::GFP, XBP1s::Turquoise dual UPR reporter, Ern1 null cell line |
| Cell line, (*Cricetulus griseus*) | CHO-K1 S21 CHOP::GFP, XBP1s::Turquoise IRE1 wild-type | This paper | IRE1 wild-type | CHO CHOP::GFP, XBP1s::Turquoise dual UPR reporter, Ern1 null cell line reconstituted with IRE1 wild-type |
| Cell line, (*Cricetulus griseus*) | CHO-K1 S21 CHOP::GFP, XBP1s::Turquoise IRE1 ΔΔ | This paper | IRE1 ΔΔ | CHO CHOP::GFP, XBP1s::Turquoise dual UPR reporter, Ern1 null cell line reconstituted with IRE1 ΔΔ (missing residues 313–338 and 391–444) |
| Peptide, recombinant protein | MPZ-N | *Karagöz et al., 2017* | MPZ-N | 12-mer peptide (MPZ-N) derived from myelin protein zero |
| Peptide, recombinant protein | FAM-MPZ-N | *Karagöz et al., 2017* | FAM-MPZ-N | FAM labelled 12-mer peptide (MPZ-N) derived from myelin protein zero |
| Software, algorithm | Prism | GraphPad | | |
| Software, algorithm | FlowJo,LLC, | | | |
| Software, algorithm | Data Analysis 4.1 | Bruker | | |
| Chemical compound, drug | Tunicamycin | Melford | Cat no: T2250 | |
| Chemical compound, drug | 2-Deoxyglucose | Sigma | Cat no: D6134 | |
| Chemical compound, drug | 4µ8c | Tocris Bioscience | Cat no: 4479 | |
| Chemical compound, drug | Digitonin | Calbiochem | Cat no: 300410 | |

*Continued on next page*

*Continued*

| Reagent type (species) or resource | Designation | Source or reference | Identifiers | Additional information |
|---|---|---|---|---|
| Chemical compound, drug | Biotin-NHS ester | Sigma | Cat no: H1759 | |
| Chemical compound, drug | Protease inhibitors | Sigma Aldrich (MERCK) | S8830 | |
| Chemical compound, drug | Oregon Green-iodoacetic acid | ThermoFisher | Cat no: O6010 | |
| Chemical compound, drug | TAMRA-maleimide | Sigma | Cat no: 94506 | |
| Chemical compound, drug | Phosphocreatine | Sigma | Cat no: 10621714001 | |
| Chemical compound, drug | Creatine kinase | Sigma | Cat no: C3755 | |
| Recombinant DNA reagent | haBiP_27–654_ pQE10 (plasmid) | *Petrova et al., 2008* | UK173 | N-terminally His6-tagged hamster BiP |
| Recombinant DNA reagent | haBiP_27–654_V461F_ pQE10 (plasmid) | *Petrova et al., 2008* | UK182 | N-terminally His6-tagged hamster BiP V461F |
| Recombinant DNA reagent | haBiP_27–654_ADDA_ pQE10 (plasmid) | *Preissler et al., 2015a* | UK984 | N-terminally His6-tagged hamster BiP ADDA |
| Recombinant DNA reagent | H6_Ulp1_pET28b (plasmid) | This study | UK1249 | $H_6$-tagged Ulp1 |
| Recombinant DNA reagent | pCEFL_mCherry_ 3XFLAG_C (plasmid) | *Sekine et al., 2016* | UK1314 | pCEFL with 3XFLAG_C tagged from mCherry-tagged plasmid |
| Recombinant DNA reagent | BPPTSP_SubA_22–347_3XFLAG _KDEL_pUC57_Acc65I_based_ pCEFL_mCherry (plasmid) | This study | UK1452 | 3xFLAG-tagged SubA with KDEL on mCherry-tagged plasmid |
| Recombinant DNA reagent | BPPTSP_SubA_22–347_S272A_ 3XFLAG_KDEL_pUC57_Acc65I_ based_pCEFL_mCherry (plasmid) | This study | UK1459 | 3xFLAG-tagged SubA[S272A] with KDEL mCherry-tagged plasmid |
| Recombinant DNA reagent | hIRE1_19–486_dC_GST_ del3UTR _pCDNA3 (plasmid) | *Amin-Wetzel et al., 2017* | UK1703 | C-GST-tagged cysteine-free human IRE1 |
| Recombinant DNA reagent | CHO_IRE1_guideC15.1_ pSpCas9(BB)−2A-mCherry (plasmid) | *Kono et al., 2017* | UK1903 | Cas9 and guide targeting IRE1 in CHO-K1 ΔLD clone 15 (mCherry-tagged) |
| Recombinant DNA reagent | CHO_IRE1_hIRE1-LD_ reptemp4_pCR-Blunt2-TOPO (plasmid) | *Kono et al., 2017* | UK1968 | Repair template for wild-type IRE1 reconstitution in CHO-K1 cells |
| Recombinant DNA reagent | Smt3_cgERdj4_24–222_ pET-21a (plasmid) | *Amin-Wetzel et al., 2017* | UK2012 | N-Smt3-tagged Chinese amster ERdj4 24–222 |
| Recombinant DNA reagent | Smt3_J4_domain_24–90_ pET-21a (plasmid) | *Amin-Wetzel et al., 2017* | UK2041 | N-Smt3-tagged Chinese hamster ERdj4 24–90 |
| Recombinant DNA reagent | pET22b_H7_Smt3_Ire1a_ LDΔC_24_444 (plasmid) | This study | UK2042 | N-His₆-Smt3-tagged wild-type human IRE1[LD]24–444 |
| Recombinant DNA reagent | pET22b_H7_Smt3_Ire1a_ LDΔC_24_444 Q105C (plasmid) | *Amin-Wetzel et al., 2017* | UK2045 | N-His₆-Smt3-tagged cysteine-free human IRE1[LD Q105C]24–444 |
| Recombinant DNA reagent | pET22b_H7_Smt3_Ire1a_ LDΔC_24_444 R234C (plasmid) | *Amin-Wetzel et al., 2017* | UK2048 | N-His₆-Smt3-tagged cysteine-free human IRE1[LD]24–444, R234C (FRET probe) |
| Recombinant DNA reagent | pET22b_H7_Smt3_Ire1a_ LDΔC_24_444 S112C (plasmid) | *Amin-Wetzel et al., 2017* | UK2076 | N-His₆-Smt3-tagged cysteine-free human IRE1[LD]24–444, S112C (FRET probe) |

*Continued*

| Reagent type (species) or resource | Designation | Source or reference | Identifiers | Additional information |
|---|---|---|---|---|
| Recombinant DNA reagent | Smt3_cgERdj4_24–222_ GS6_MalE_pET21a (plasmid) | *Amin-Wetzel et al., 2017* | UK2108 | N-Smt3-ERdj4-MBP Chinese hamster 24–222 |
| Recombinant DNA reagent | Smt3_cgERdj4_24–222_ QPD_GS6_MalE_pET21a (plasmid) | *Amin-Wetzel et al., 2017* | UK2119 | N-Smt3-ERdj4-MBP Chinese hamster residues 24–222 H54Q |
| Recombinant DNA reagent | IRE1a_LD_ΔC_24–443_ AviTag_H6_pET30a (plasmid) | This study | UK2246 | C-Avi-His$_6$-tagged cysteine-free human IRE1$^{LD}$24–444 |
| Recombinant DNA reagent | pET22b_H7_Smt3_ Ire1a_LDΔC_Q105C_ 24_390 (plasmid) | This study | UK2304 | N-His$_6$-Smt3-tagged cysteine-free human IRE1$^{LD\ Q105C}$24–390 |
| Recombinant DNA reagent | pET22b_H7_Smt3_Ire1a _LD_dC_24_390_Δ313–338_ S112C (plasmid) | This study | UK2370 | N-His$_6$-Smt3-tagged cysteine-free human IRE1$^{LD\ \Delta\Delta}$ (313-338, 391-444) S112C, FRET probe |
| Recombinant DNA reagent | CHO_IRE1_hIRE1-LD_d313 -338_reptemp4_pCR- Blunt2-TOPO (plasmid) | This study | UK2384 | Repair template for IRE1 Δloop (d313-338) reconstitution in CHO-K1 cells |
| Recombinant DNA reagent | CHO_IRE1_hIRE1-LD_d391- 444_reptemp4_pCR-Blunt2- TOPO (plasmid) | This study | UK2385 | Repair template for IRE1 Δtail (d391-444) reconstitution in CHO-K1 cells |
| Recombinant DNA reagent | CHO_IRE1_hIRE1-LD_d313- 338_d391-440_reptemp4_ pCR-Blunt2-TOPO (plasmid) | This study | UK2386 | Repair template for IRE1 ΔΔ (d313-338, 391–444) reconstitution in CHO-K1 cells |
| Recombinant DNA reagent | hIRE1α_19–486_dC_ d313-338 _d391-440_GST_del3UTR _ pCDNA3 (plasmid) | This study | UK2401 | C-GST-tagged cysteine-free human IRE1 ΔΔ (missing residues 313–338 and 391–444) |
| Recombinant DNA reagent | hIRE1α_19–486_dC_ d313-338 _GST_del3UTR _pCDNA3 (plasmid) | This study | UK2404 | C-GST-tagged cysteine-free human IRE1 Δloop (missing residues 313–338) |
| Recombinant DNA reagent | hIRE1α_19–486_dC_ d391-440 _GST_del3UTR _pCDNA3 (plasmid) | This study | UK2406 | C-GST-tagged cysteine-free human IRE1 ΔΔ (missing residues 391–444) |
| Recombinant DNA reagent | Met_ERdj4_24–120_Ire1a_LDΔC_ 24–443_AviTag_H6_pET30a (plasmid) | This study | UK2408 | C-Avi-His$_6$-tagged cysteine-free chimeric J-ERdj4 human IRE1$^{LD}$24–444 protein |
| Recombinant DNA reagent | pET22b_H7_Smt3_Ire1a_ LD_dC_24_444_P108A (plasmid) | This study | UK2410 | N-His$_6$-Smt3-tagged cysteine-free human IRE1$^{LD\ P108A}$ monomeric mutant 24–444 |
| Recombinant DNA reagent | pET22b_H7_Smt3_Ire1a_ LD_dC_24_444_W125A (plasmid) | This study | UK2411 | N-His$_6$-Smt3-tagged cysteine-free human IRE1$^{LD\ W125A}$ monomeric mutant 24–444 |
| Recombinant DNA reagent | Met_ERdj4_24–120_ Ire1a_LDΔC_24–443_S112C_ AviTag_H6_pET30a (plasmid) | This study | UK2412 | C-Avi-His$_6$-tagged cysteine-free chimeric J-ERdj4 human IRE1$^{LD}$24–444 protein, S112C (FRET probe) |
| Recombinant DNA reagent | J4_WT_IRE1_LD_ CHORepairTemplate (plasmid) | This study | UK2425 | Repair template for chimeric J-IRE1 reconstitution in CHO-K1 cells |
| Recombinant DNA reagent | J4_QPD_IRE1_LD_ CHORepairTemplate_ V1 (plasmid) | This study | UK2426 | Repair template for chimeric J$^{QPD}$-IRE1 reconstitution in CHO-K1 cells |

*Continued on next page*

*Continued*

| Reagent type (species) or resource | Designation | Source or reference | Identifiers | Additional information |
|---|---|---|---|---|
| Recombinant DNA reagent | Met_ERdJ4_24–120_ Ire1a_LDΔC_24–443_P108A_ AviTag_H6_pET30a (plasmid) | This study | UK2428 | C-Avi-His$_6$-tagged cysteine-free chimeric J-ERdj4 human IRE1$^{LD}$24–444 protein containing monomerising mutation P108A |
| Recombinant DNA reagent | Met_ErdJ4_24–120_ IRE1a_LDΔC_24–390_Δ313–338_ AviTag_H6_pET30a (plasmid) | This study | UK2458 | C-Avi-His$_6$-tagged cysteine-free chimeric J-ERdj4 human IRE1$^{LD\ \Delta\Delta}$ protein (313-338, 391-444) |
| Recombinant DNA reagent | IRE1a_LDΔC _24–390_Δ313–338_AviTag_H6 _pET30a (plasmid) | This study | UK2459 | C-Avi-His$_6$-tagged cysteine-free human IRE1$^{LD\ \Delta\Delta}$ (d313-338, 391–444) |
| Recombinant DNA reagent | Met_ERdJ4_24–120_ Ire1a_LDΔC_24–443_Q105C_ AviTag_H6_pET30a (plasmid) | This study | UK2558 | C-Avi-His$_6$-tagged cysteine-free chimeric J-ERdj4 human IRE1$^{LD}$24–444 protein containing mutation Q105C |

## Mammalian cell culture

The parental strains for the CRISPR-Cas9-mediated homologous recombination approaches were the previously described ΔLD15 dual CHOP::GFP and XBP1s::Turquoise UPR reporter Chinese Hamster Ovary CHO-K1 cell lines (*Kono et al., 2017*) and have been authenticated as CHO-K1 using the criteria of successful targeting of essential genes using a species-specific CRISPR whole genome library, and sequencing of the wild-type or mutant alleles of the genes studied that confirmed the sequence reported for the corresponding genome. The cell lines have tested negative for mycoplasma contamination using a commercial kit (MycoAlert (TM) Mycoplasma Detection Kit, Lonza). None of the cell lines is on the list of commonly misidentified cell lines maintained by the International Cell Line Authentication Committee. The CRISPR-Cas9-mediated mutagenesis strategy and the transient transfection of GST-tagged IRE1$^{LD}$ was performed with CHO-K1 S21 dual UPR reporter cells (*Sekine et al., 2016*). Cells were cultured in Ham's nutrient mixture F12 (Sigma). All cell media was supplemented with 10% (v/v) serum (FetalClone-2, Hyclone), 2 mM L-glutamine (Sigma), 100 U/ml penicillin and 100 µg/ml streptomycin (Sigma). Cells were grown in tissue culture dishes or multi-well plates (Corning) at 37°C and 5% $CO_2$. Tunicamycin (Melford) treatment was at 2.5 µg/ml for 16 hr, 2-Deoxyglucose (2DG) (Sigma) treatment at 4 mM for 16 hr and 4µ8c (*Cross et al., 2012*) treatment at 10 µM for 7 days. The drugs were mixed with pre-warmed culture medium and immediately added to the cells by medium exchange.

## Transfection

Cells were transfected using Lipofectamine LTX (Life Technologies) transfection reagent with reduced serum medium Opti-MEM (Life Technologies) following the manufacturer's instructions.

## Flow cytometry and fluorescence-activated cell sorting (FACS)

To analyse the effect of IRE1 variants expressed from the endogenous *Ern1* locus on the UPR (*Figure 1A and B*, *Figure 4—figure supplement 1A and B*), flow cytometry was performed. Cells were washed once in PBS and collected in PBS containing 4 mM EDTA. Single-cell fluorescent signals (20,000/sample) were analysed by dual-channel flow cytometry with an LSRFortessa cell analyser (BD Biosciences). FACS was performed on either a Beckman Coulter MoFlo or a BD FACSMelody cell sorter. Cells were washed once in PBS and then incubated 5 min in PBS supplemented with 0.5% BSA and 4 mM EDTA before sorting into fresh media. CHOP::GFP fluorescence was detected with excitation laser at 488 nm, filter 530/30 nm; XBP1s::Turquoise fluorescence with excitation laser 405 nm, filter 450/50 nm and mCherry fluorescence with excitation laser 561, filter 610/20. To generate clonal cell lines stably expressing a version of IRE1 the transfected cells were treated with 2-Deoxyglucose to gate for cells showing high CHOP::GFP XBP1s::Turquoise fluorescence.

## Gene manipulation and allele analysis

Cas9 guides were either manually designed following standard guidelines (*Ran et al., 2013*) or taken from the CRISPy database (URL: http://staff.biosustain.dtu.dk/laeb/crispy/, (*Ronda et al., 2014*). Cells were transfected with the Cas9 and guide constructs and grown for seven days before they were analysed by flow cytometry or FACS.

For the in vivo mutagenesis strategy (*Figure 3B and C* and *Figure 3—figure supplement 1B*), a series of guides that tiled the two regions of interest, set A covering the putative loop (residues 308–362) and set B covering the tail (residues 368–444) was designed. Set A and set B guide-Cas9 encoding plasmids were transfected singly or in different pairwise combinations into IRE1 wild-type expressing cells (CHO-K1 S21 CHOP::GFP, XBP1s::Turquoise dual reporter cell line) and pooled to create population 0 (*Figure 3C*). Rare de-repressing IRE1 mutants were enriched from the mutagenised population by iterative rounds of FACS (populations 1 and 2) followed by a selection against clones that had acquired IRE1-independent XBP1s::Turquoise reporter expression, as assessed by their unresponsiveness to the IRE1 inhibitor 4µ8c. Genomic DNA was extracted from final clones, PCR used to amplify the loci of interest and the resultant products were sequenced. The genomic DNA was extracted from cells by incubation in Proteinase K solution (100 mM Tris-HCl pH 8.5, 5 mM EDTA, 200 mM NaCl, 0.25% SDS, 0.2 mg/ml Proteinase K) overnight at 50 ˚C. Next, Proteinase K was heat inactivated at 98 ˚C for 20 min before the supernatant was collected and used as a template in PCR reactions before sequencing. To facilitate the interpretation of the sequencing data, the changes in size of alleles modified by Cas9 was determined by capillary electrophoresis on a 3730xl DNA analyser (Applied Biosystems). For that, sample preparation was performed with one of the oligonucleotides having a 5' 6-carboxyfluorescein (FAM) flurophore in the PCR reaction.

## Creating clonal cell lines stably expressing IRE1 variants

The activity of IRE1 variants was analysed by introducing them into the endogenous *Ern1* locus of CHO-K1 CHOP::GFP and XBP1s::Turquoise dual UPR reporter cells using a *Ern1* null cell line (ΔIRE1 as described in *Kono et al., 2017*). Cells were transfected with a Cas9-CRISPR guide construct targeting the *Ern1* locus (UK1903) together with the respective repair templates (UK2425 for chimeric J-IRE1, UK2426 for J$^{QPD}$-IRE1, UK1968 for wild-type IRE1, UK2384 for IRE1 Δloop, UK2385 for IRE1 Δtail, UK2386 for IRE1 ΔΔ) and grown for 7 days before further analysis. Cells that successfully repaired the IRE1 locus were selected by FACS by gating for cells exhibiting increased XBP1s::Turquoise fluorescence after 2-deoxyglucose treatment. Cells transfected with J-IRE1 as repair template were additionally transiently transfected with a plasmid encoding SubA wild-type, mutant or an empty vector (UK1452, UK1459, UK1314 respectively) before FACS. Data shown in *Figure 4A* and *Figure 4—figure supplement 1A* was acquired after transient transfection using a mixed population of cells and data shown in *Figures 1A, B, C* and *4B* and *Figure 4—figure supplement 1B* with clonal cell lines.

## Mammalian cell lysis

Cell lysis was performed as described previously (*Amin-Wetzel et al., 2017*). All reagents were kept on ice throughout. Cells were washed in PBS, removed from the culture dish in PBS + 1 mM EDTA with a cell scraper and then pelleted at 370 × *g* for 5 min at 4˚C. Cells were incubated in lysis buffer (1% Triton X-100, 150 mM NaCl, 20 mM HEPES-KOH pH 7.5, 10% glycerol, 1 mM phenylmethylsulphonyl fluoride (PMSF), 4 µg/m Aprotinin, 2 µg/ml Pepstatin A and 2 µM Leupeptin) for 5 min. Next, the samples were clarified at 21,130 g for 10 min at 4˚C. The supernatant was transferred to a fresh tube and protein concentration measured with BioRad protein assay reagent (Bio-Rad).

For BiP co-IP experiments, non-specific binding of BiP to protein-A sepharose beads was decreased by digitonin treatment (*Le Gall et al., 2004*) to remove non-membrane associated BiP from cells prior to lysis. After pelleting, cells were washed in HNC buffer (50 mM HEPES-KOH pH 7.5, 150 mM NaCl, 2 mM CaCl$_2$) and then incubated in HNC + 0.1% (w/v) digitonin (Calbiochem) for 10 min. Cells were then washed in HNE buffer (50 mM HEPES-KOH pH 7.5, 150 mM NaCl, 1 mM EGTA) before proceeding to lysis using lysis buffer supplemented with 10 mM MgCl$_2$, 6 mg/ml glucose and 50 U/ml Hexokinase (H4502 Sigma) to deplete ATP and stabilise BiP-substrate interactions.

## Immunoprecipitation (IP) and GST pull down assays

To analyse the amount of BiP co-immunoprecipitated with endogenously expressed IRE1 variants (*Figures 1C* and *4B*) or transiently transfected IRE1$^{LD}$-GST variants (*Figure 4C*), Protein A sepharose 4B beads (Zymed Invitrogen) or Glutathione (GSH) Sepharose 4B beads (GE Healthcare) were equilibrated in lysis buffer. Next, 20 µl beads per sample and anti-IRE1α were added to lysates and left rotating for 1 hr at 4°C. The beads were then washed in lysis buffer and residual liquid removed using a syringe. The protein from the beads was eluted in SDS sample buffer containing 20 mM DTT.

## Antibodies

Anti-mouse IRE1α serum (NY200) was used for IP and immunoblot detection of endogenous IRE1α (*Bertolotti et al., 2000*). An anti-hamster BiP antibody was used for immunoblot detection of endogenous BiP (*Avezov et al., 2013*). Anti-GST serum was used for immunoblot detection of GST fusion proteins (*Ron and Habener, 1992*).

## Reducing/non-reducing SDS-PAGE and immunoblotting

Samples were run on standard polyacrylamide Tris-HCl gels and transferred to Immobilon-P PVDF membrane (Pore size 0.45 µm, Sigma). Membranes were then blocked in 5% (w/v) dried skimmed milk in PBS, washed in TBS with 0.1% Tween-20 and exposed to various primary antibodies/antisera followed by incubation with IRDye fluorescently labelled secondary antibodies. Imaging was carried out with using a LICOR CLx Odyssey infrared imager. Coomassie-staining was carried out with Instant Blue (Expedeon). Signal quantitation from SDS-PAGE gels or from immunoblots was carried out using the ImageJ software (NIH).

## Protein purification

### Human IRE1$^{LD}$

His$_6$-Smt3-IRE1$^{LD}$ (UK2048, UK2079, UK2042, UK2370, UK2410, UK2411, UK2516, UK2045, UK2304) and His$_6$-Avitag-IRE1$^{LD}$ (UK2412, UK2246, UK2459, UK2408, UK2458) variants were encoded on a pET-derived vector (Novagen) as fusion proteins and expressed in T7 Express *lysY/I$^q$ E. coli* cells (NEB). IRE1$^{LD\ Q105C}$ variants (UK2045, UK2304) used to make disulfide-linked dimeric IRE1$^{LD\ Q105C}$ species were expressed in Origami B(DE3) cells (Novagen).

Protein purification was performed as described in *Amin-Wetzel et al. (2017)*. Bacterial cultures were grown at 37°C in LB medium containing 100 mg/ml ampicillin until an OD$_{600nm}$ of 0.6–0.8 was reached. Expression was induced with 0.5 mM IPTG and the cells were incubated for 16 hr at 18°C. After sedimentation of the cells by centrifugation the pellets were resuspended in TNGM buffer (50 mM Tris-HCl pH 7.4, 500 mM NaCl, 10% glycerol, 1 mM MgCl$_2$). The cell suspension was supplemented with 0.1 mg/ml DNaseI and protease inhibitors (2 mM PMSF, 4 mg/ml Pepstatin, 4 mg/ml Leupeptin, 8 mg/ml Aprotinin) and lysed by repeated passage through a high-pressure homogenizer (EmulsiFlex-C3, Avestin). After clarification of the lysates by centrifugation at 20,000 × *g* for 30 min the supernatant was removed and incubated for 60 min at 4°C with Ni-NTA agarose (Qiagen) (0.5 ml per liter of bacterial culture). The matrix was washed two times with 50 ml of TNGMI wash buffer (50 mM Tris-HCl pH 7.4, 500 mM NaCl, 10% glycerol, 1 mM MgCl$_2$, 20 mM imidazole). The matrix was transferred to a gravity-flow column and the flow-through was collected after a wash with one bed volume of elution buffer (50 mM Tris-HCl pH 7.4, 100 mM NaCl, 10% glycerol, 250 mM imidazole). The protein solutions were concentrated using 30 kDa MWCO centrifugal filters (Amicon Ultra; Merck Millipore), flash frozen and stored at −80 °C.

For the preparation of UK2042, UK2045, UK2410, UK2411 and UK2516 1.5 µg/ml His$_6$-Ulp1 (UK1249) and 1 mM TCEP were added to the eluates and incubated overnight at 4°C, whilst being dialysed against HKM buffer (50 mM HEPES-KOH pH 7.4, 150 mM KCl, 10 mM MgCl$_2$). To remove the cleaved His$_6$-Smt3-tag and the His$_6$-Ulp1 the solution was again incubated with Ni-NTA agarose for 60 min at 4°C. After passing the sample through a gravity-flow column the final eluate was collected, concentrated using 30 kDa MWCO centrifugal filters, flash frozen and stored at −80 °C.

For the preparation of UK2304, the dialysis overnight was performed against TN buffer (150 mM NaCl, 50 mM Tris-HCl pH 7.4). After removal of uncleaved protein and His$_6$-Smt3 the protein solution was then diluted to 75 mM NaCl, 50 mM Tris-HCl pH 7.4, 10 mM imidazole and bound to an

anion exchange column. The protein was eluted in 10 mM Tris-HCl pH 7.4 50–500 mM NaCl and then incubated with 5 mM oxidised glutathione overnight. The sample was then separated on a Superdex 200 10/300 GL gel filtration column equilibrated in TN buffer and appropriate fractions collected and concentrated using 30 kDa MWCO centrifugal filters, flash frozen and stored at −80 °C.

The purification of the FRET probes (UK2048, UK2076, UK2412, UK2370) was performed as described above with 1 mM TCEP contained in all buffers. Eluted fractions were buffer exchanged into HKMT buffer (50 mM HEPES-KOH pH 7.4, 150 mM KCl, 10 mM MgCl$_2$, 1 mM TCEP) using a CentiPure P10 desalting column (Generon) and labelled with threefold molar excess of Oregon Green-iodoacetic acid (ThermoFisher) or TAMRA-maleimide (Sigma) to make IRE1$^{LD}$ $^{R234C}$-OG (UK2048), IRE1$^{LD}$ $^{R112C}$-TMR (UK2076), J-IRE1$^{S112C}$-TMR (UK2412) and IRE1$^{LD}$ $^{ΔΔ}$ $^{S112C}$-TMR (UK2370). The reaction proceeded at room temperature in the dark overnight and was quenched by the addition of 5 mM DTT. The reaction mixture was passed through a CentiPure P10 gravity-desalting column (Generon) equilibrated in HKM buffer and afterwards through a Superdex 200 10/300 GL gel filtration column equilibrated in HKG (50 mM HEPES-KOH pH 7.4, 150 mM KCl, 10% (v/v) glycerol) buffer. Appropriate fractions were collected, concentrated, flash frozen and stored at −80 °C.

For the streptavidin pull down (*Figure 1D* and *Figure 5B*) and the Bio-Layer Interferometry (BLI) measurements (*Figure 5A* and *Figure 5—figure supplement 1A and B*), chimeric J-IRE1$^{LD}$ (UK2408), J-IRE1$^{LD}$ $^{ΔΔ}$ (UK2458), J-IRE1$^{LD}$ $^{P108A}$ (UK2428) and J-IRE1$^{LD}$ $^{Q105C}$ $^{SS}$(UK2558), wild-type IRE1$^{LD}$ (UK2246) and IRE1$^{LD}$ $^{ΔΔ}$ (UK2459) were biotinylated enzymatically with *E. coli* BirA to create biotinylated J-IRE1$^{LD}$-bio and IRE1$^{LD}$-bio variants, respectively.

## Hamster ERdj4

Expression and purification of ERdj4 and variants was performed according to the protocol previously described in *Amin-Wetzel et al. (2017)*. The constructs were expressed as fusion proteins with an N-terminal His$_6$-Smt3 (UK2012 for wild-type ERdj4) or with both, an N-terminal His$_6$-Smt3 and C-terminal MBP (UK2108 for wild-type and UK2119 for QPD ERdj4) in Origami B(DE3) cells. Cells were grown and lysed as described above for His$_6$-Smt3 tagged proteins. Ni-NTA chromatography was performed as described above. His$_6$-Smt3-ERdj4-MBP variants were further purified on a S200 10/300 GL column equilibrated in HKM buffer.

## Hamster BiP

BiP and BiP variants (UK173, UK182, UK984) were purified as previously described in *Petrova et al. (2008)*; *Preissler et al. (2015a)*.

## Streptavidin pull down assays

To assess BiP binding to IRE1$^{LD}$ variants in the presence and absence of J-domain-mediated hyper-affinity (*Figure 1D* and *Figure 5B*, respectively), 20 µl Dynabeads MyOne Streptavidin C1 (Thermo Fisher Scientific) per sample were used. Analysis was performed in HKMGTw buffer (50 mM HEPES-KOH pH 7.4, 150 mM KCl, 10 mM MgCl$_2$, 10% (v/v) glycerol, 0.05% TWEEN-20). Reactions contained 5 µM biotinylated IRE1$^{LD}$ proteins (UK2246, UK2408, UK2459), 8 µM ERdj4 (UK2012), 30 µM BiP variants (UK173, UK182, UK984), and 2 mM ATP. In experiments conducted in presence of a J-domain the samples were incubated for 20 min at 30°C. In experiment conducted in absence of a J-domain for 16 hr at 4°C. Next, the samples were clarified at 21,130 × *g* for 5 min and an excess of ice cold 1 mM ADP was added to the supernatant followed by the addition of Dynabeads. Binding was performed for 45 min at 4°C followed by washing in assay buffer supplemented with 1 mM ADP (for the ATP wash 2 mM ATP was used instead). The samples were eluted with 1x SDS sample buffer.

## Analytical size-exclusion chromatography (SEC)

To assess the oligomeric state of wild-type IRE1$^{LD}$ (UK2042), IRE1$^{LD}$ $^{W125A}$ (UK2411) and IRE1$^{LD}$ $^{P108A}$ (UK2410) and IRE1$^{LD}$ $^{ΔΔ}$ (UK2516) in presence and absence of MPZ-N peptide (GeneScript, Piscataway, NJ), SEC was performed (*Figure 2A and B*, *Figure 2—figure supplement 1A* and *Figure 5—figure supplement 1C*). Samples were run through a SEC-3 HPLC column (300 Å pore size; Agilent Technologies) on an Agilent infinity HPLC system equilibrated in HKM buffer at a flow rate of 0.3 ml/

min. Samples were pre-incubated in a final volume of 20 µl for 30 min at 30°C before clarification at 21,130 × g for 5 min and subsequent injection of 10 µl. Runs were performed at 25°C and $A_{280}$ absorbance and TAMRA (TMR, excitation 544 nm and emission 572 nm) traces were recorded.

## Bio-layer interferometry (BLI) experiments

Experiments were performed on an Octet RED96 (Pall ForteBio) in HKM buffer supplemented with 0.05% Triton X-100. In the sequential dipping experiments (*Figure 5A* and *Figure 5—figure supplement 1A and B*), streptavidin biosensors were loaded with the indicated biotinylated IRE1$^{LD}$ variants (UK2246, UK2428, UK2558, UK2459, UK2408, UK2458) to approximately 1 nm displacement, washed in assay buffer, and then sequentially dipped in wells containing the indicated analytes. For the two-component system (*Figure 5A* and *Figure 5—figure supplement 1B*) with the chimeric J-IRE1$^{LD}$ fusions 0.15 or 0.5 µM BiP wild-type (UK173), 0.5 µM BiP$^{V461F}$ (UK182) and 2 mM ATP were used and for the three-protein system (*Figure 5—figure supplement 1A*) 2 µM BiP (UK172), 6.8 µM ERdj4 full-length (UK2012) or J-domain (UK2041) and 2 mM ATP. After each association, the sensor was dipped into buffer containing 2 mM ATP to allow for BiP dissociation. Data were normalised to the signal after the first wash step.

## Kinetic FRET experiments

To assess the effect of a fused J-domain to IRE1$^{LD}$ (*Figure 1E*) or the introduction of deletions into the IRE1$^{LD}$ on Erdj4 and BiP-mediated monomerisation (*Figure 5C*) kinetic FRET measurements were performed. For this, heterodimeric FRET pairs consisting of an OG-labelled wild-type donor molecule combined with a TMR-labelled mutant acceptor molecule were employed. Although this experimental setup reduces the dynamic range of the measurements, it is compensated by the greater comparability of detecting changes in donor fluorescence in presence of a constant IRE1$^{LD}$-OG donor molecule. IRE1$^{LD}$-OG donor (UK2048-OG) and IRE1$^{LD}$-TMR acceptor (UK2076-TMR, UK2412-TMR, or UK2370-TMR) molecules were combined at a 1:2 ratio and incubated at room temperature in the dark for two hours. In *Figure 1E* 30 µM BiP, 2.5 µM ERdj4 (UK2108), and 0.2 µM IRE1$^{LD}$-FRET pair were combined in HKMGTw buffer and incubated for 30 min. To initiate the reaction, 2 mM ATP with an ATP regeneration system (8 mM phosphocreatine, 0.016 mg/ml creatine kinase) was added. In *Figure 5C*, 30 µM BiP, 2.5 µM ERdj4 full-length (UK2108) or J-ERdj4 (UK2041), and 0.2 µM pre-equilibrated IRE1$^{LD}$-FRET pair were combined in HKMGTw buffer. The donor fluorescence was followed with a CLARIOstar plate reader (excitation: 470–15 nm, emission: 524–20 nm) recording signals every 30 s. The donor fluorescence was background subtracted arising from a well containing buffer only and analysed with the Prism GraphPad 7.0 software.

## Differential scanning fluorimetry (DSF)

DSF experiments were performed on a CFX96 Real-Time System (Bio-Rad). Reactions were transferred into 96-well qPCR plates (Thermofisher) (final volume 25 µl). Protein concentrations were 5 µM, ligands at the concentration indicated in the figure (62.5–1000 µM), and SYPRO Orange (Thermofisher) dye at a 10x concentration in HKM buffer. Where indicated 1 mM TCEP was included. Over a temperature range of 20–95°C fluorescence of the SYPRO Orange dye was monitored using the SYBR-FAM filter set. Data was then analysed in Prism GraphPad 7.0, with melting temperature calculated as the global minimum of the negative first derivative of the respective fluorescent unit melt curves. Data shown in *Figure 2—figure supplement 1D* indicate a correlation between the monomer-dimer equilibrium and the $T_m$ of IRE1$^{LD}$. In line with that, titrational analysis of IRE1$^{LD}$ and variants reported on an increased $T_m$ at higher protein concentrations, however it did not result in a two state transition representing the monomer and the dimer because of oligomerisation (data not shown). Moreover, the lower $T_m$ of the monomeric variants could reflect an intrinsic destabilisation of the protein caused by the point mutation.

## Fluorescence polarisation (FP)

To characterise the binding of FAM labelled MPZ-N peptide [5-FAM-LIRYCWLRRQAA) (as described in *Karagöz et al. (2017)*, from GeneScript, Piscatawy, NJ] to IRE1$^{LD}$ or disulphide-linked IRE1L$^{D}_{Q105C\ SS}$. (*Figure 2D*). FP was measured with a CLARIOstar plate reader. Using excitation at 496 nm and measuring emission at 519–550 nm, parallel and perpendicular fluorescence of the FAM

fluorophore was detected. Whilst FAM-MPZN was kept at 100 nM in reactions the concentrations of IRE1$^{LD}$ variants are detailed in the legend of *Figure 2D*. Samples containing the respective components were prepared in 25 µl and then 20 µl were transferred to a black flat-bottomed 384 well plate and incubated for 30 min prior to reading. Fluorescence readings were corrected by subtracting fluorescence from a well containing only buffer. The average of three readings (spaced at 30 s intervals) per well was taken as one repeat and the average of three independent repeats was used for *Figure 2D*. Anisotropy was calculated according to *Equation 1*:

$$A = \frac{I_{para} - I_{perp}}{I_{para} + 2I_{perp}} \tag{1}$$

The data was fit to *Equation 2*:

$$r_{free} + \frac{(r_{max} - r_{free})^h}{[X]^h + K^h} \tag{2}$$

whereby $r_{free}$ represents anisotropy in the absence of protein, $r_{max}$ the theoretical maximal anisotropy, [X] the protein concentration and h the hill-coefficient. Curve fitting was per- formed with Prism GraphPad 7.0 and minimal constraints to obtain $K_{1/2max}$ values with $R^2$ values > 0.9. As this equation does not consider the equilibria between IRE1$^{LD}$ dimers and oligomers, the $K_{1/2max}$ value does not reflect the dissociation constant.

## Hydrogen exchange mass spectrometry (HX-MS)

For *Figure 6A* and *Figure 6—figure supplement 1A*, 5 µM wild-type IRE1$^{LD}$ (UK2042) or monomeric IRE1$^{LD\ W125A}$ (UK2411) and IRE1$^{LD\ P108A}$ (UK2410) were pre-incubated for 30 min at 30 ˚C in HKM buffer. The samples were then diluted 1:20 in D$_2$O buffer supplemented with 10 mM ADP (HKM buffer was lyophilised and re-dissolved in pure D$_2$O at least three times) and incubated for 30 and 300 s at 30˚C. Deuterated samples were quenched 1:1 with ice-cold quench buffer (2% formic acid), immediately subjected to LC-MS using an Agilent UPLC and a MaXis mass spectrometer (Bruker). For each experiment at least one unexchanged sample and one fully deuterated control were measured. The unexchanged protein sample was diluted 1:20 in H$_2$O buffer. The fully deuterated sample (protein in HKM buffer containing 6 M guanidine hydrochloride, lyophilised and re-dissolved in pure D$_2$O at least three times) was treated as the other samples. For HX-MS experiments presented in *Figure 6C*, *Figure 6—figure supplements 1B, C*, 5 µM wild-type IRE1$^{LD}$ (UK2042) or IRE1$^{LD\ \Delta\Delta}$ (UK2516) was incubated with 30 µM BiP wild-type (UK173) or V461F mutant (UK182), 6 µM ERdj4 wild-type (UK2108) or ERdj4 QPD mutant (UK2119) and 2 mM ATP in HKM buffer. Dilution in D$_2$O buffer and subsequent steps were performed as described above. Data analysis was performed manually (Data Analysis 4.1, Bruker).

## HX-MS data analysis of the bimodal distributed isotope clusters

In order to observe an amide hydrogen exchange, a structure-specific H-bond between a peptide backbone amide hydrogen and an H-bond acceptor has to open. This opening occurs by unfolding of secondary structures within the native conformation of the protein according to *Equation 3*:

$$\mathbf{F(H)} \underset{k_{cl}}{\overset{k_{op}}{\rightleftarrows}} \mathbf{U(H)} \rightarrow [\mathrm{OD}^-/\mathrm{D_2O}]k_{ch}\mathbf{U(D)} \underset{k_{op}}{\overset{k_{cl}}{\rightleftarrows}} \mathbf{F(D)} \tag{3}$$

Hereby, F and U indicate the folded and unfolded state of the structural element, respectively. The conversions between these conformations are determined by $k_{op}$ and $k_{cl}$, representing the opening and closing rate constants. The intrinsic chemical exchange rate is indicated by $k_{ch}$.

There are two extreme cases, distinguishing the so-called EX1 and EX2 exchange regimes. In case of EX1 $k_{cl}$ is much smaller than $k_{ch}$. Therefore, all amide protons exchange at once upon unfolding and the observed rate is practically equal to the opening rate $k_{op}$ of the structural element. This type of exchange kinetics is characterised by a bimodal distribution of the isotope peaks in peptic mass spectra showing two separate, interconverting subpopulations (*Figure 7—figure supplement 1A*). Notably, the EX1 exchange regime occurs only rarely under native conditions and is more likely to be observed in presence of chemical denaturants as they decrease the closing rate $k_{cl}$ without

affecting the intrinsic chemical exchange rate $k_{ch}$ by interfering with H-bond and hydrophobic core formation. EX2 is most commonly observed, as under native state conditions $k_{cl}$ is generally much greater than the intrinsic chemical exchange rate $k_{ch}$. Hence, many opening and closing cycles are necessary in order to exchange all amide protons to deuterons, which is visible by a gradual increase of the average mass in the peptic mass spectra whilst the isotopic distribution remains roughly the same (*Rist et al., 2003*). By comparing IRE1$^{LD}$ with the monomeric variants we identified regions exhibiting increased deuteron incorporation in the monomeric state of the protein, with some of them revealing bimodal distributions of the isotope clusters. In order to calculate the contribution of each subpopulation to the peak intensities, an equation for two Gaussian distributions was fitted to the isotope peak maxima versus m/z plots using the Prism GraphPad 7.0 software:

$$I = \frac{A_1}{\sigma \cdot \sqrt{2\pi}} \cdot e^{-\frac{1}{2}\left(\frac{\mu - \bar{\mu_1}}{\sigma}\right)^2} + \frac{A_2}{\sigma \cdot \sqrt{2\pi}} \cdot e^{-\frac{1}{2}\left(\frac{\mu - \bar{\mu_2}}{\sigma}\right)^2} \tag{4}$$

Hereby, $A_{1/2}$ represents the area of the two peaks; $\mu$, the m/z values; −, the means of the Gaussian distributions, representing the centroid of each of the two subpopulations; and $\sigma$, the standard deviation of the Gaussian distributions, corresponding to the width of the isotope peak distribution. For each peptide exhibiting a bimodal distribution all intensity values belonging to one incubation time in $D_2O$ were globally fitted assuming that $\sigma$ and $\mu_{1/2}$ are constant. Independent experiments were treated independently. Next, the fitted parameters $A_{1/2}$, $\mu_{1/2}$ and $\sigma$ were used to calculate the proportion of the low and high mass subpopulation for each individual isotope peak.

## Crystallisation, data collection and structure determination

Initial crystals were obtained by screening commercial crystallisation plates via 200 nl protein (16 mg/ml) plus 200 nl well solution in 96-well sitting drop plates at 20°C. The best diffraction dataset was collected from a crystal grown in 9% MPD, 0.1 M HEPES-KOH pH 7.5 microseeded (*D'Arcy et al., 2007*) from diluted initial crystals in 20% MPD, 0.1 M HEPES-KOH pH 7.5. Crystals were briefly soaked into 9% MPD, 0.1 M HEPES-KOH pH 7.5, 25% (v/v) glycerol and cryocooled in liquid nitrogen. Diffraction data was collected at beamline I04-1 in the Diamond Synchrotron Light Source (DLS) and processed by the XIA2 pipeline (*Winter, 2010*) implementing Dials (*Winter et al., 2018*) for indexing and integration, Pointless for space group determination, and Aimless for scaling and merging (*Evans, 2011*). The structure was solved by searching the published IRE1$^{LD}$ core structure (PDB 2HZ6) using Phaser (*McCoy et al., 2007*). One molecule of IRE1 was found in one asymmetric unit, but the electron density around Cys105 and the SG-SG bond length suggested that Cys105 formed a disulphide bond with the symmetric Cys105. Further refinement was performed iteratively using COOT (*Emsley et al., 2010*) and refmac5 (*Winn et al., 2001*) (*Table 1*) in *CCP4i2* interface (*Potterton et al., 2018*) and phenix.refine (*Liebschner et al., 2019*). MolProbity (*Chen et al., 2010*) was consulted throughout the refinement process. Molecular graphics were generated with PyMOL Molecular Graphics System, Educational-use-only version 4.5 Schrodinger, LLC).

## Acknowledgements

We thank the CIMR flow cytometry core facility team (Reiner Schulte, Chiara Cossetti and Gabriela Grondys-Kotarba) for assistance with FACS, the Huntington lab for access to the Octet machine, Steffen Preissler for advice on data interpretation, Roman Kityk and Nicole Luebbehusen for help and advice with HX-MS experiments.

## Additional information

### Competing interests

David Ron: Senior editor, *eLife*. The other authors declare that no competing interests exist.

## Funding

| Funder | Grant reference number | Author |
|---|---|---|
| Medical Research Council | | Niko Amin-Wetzel |
| European Molecular Biology Organization | | Lisa Neidhardt |
| Deutsche Forschungsgemeinschaft | SFB1036 TP9 | Matthias P Mayer |
| Wellcome | Wellcome 996 100140 | David Ron |
| Wellcome | Wellcome 200848/Z/16/Z | David Ron |
| Deutsche Forschungsgemeinschaft | MA 1278/4-3 | Matthias P Mayer |

The funders had no role in study design, data collection and interpretation, or the decision to submit the work for publication.

## Author contributions

Niko Amin-Wetzel, Conceptualization, Data curation, Validation, Visualization, Methodology; Lisa Neidhardt, Data curation, Formal analysis, Validation, Visualization; Yahui Yan, Data curation, Visualization; Matthias P Mayer, Formal analysis, Supervision; David Ron, Conceptualization, Supervision, Funding acquisition, Methodology, Project administration

## Author ORCIDs

Niko Amin-Wetzel  https://orcid.org/0000-0002-4640-3724
Lisa Neidhardt  https://orcid.org/0000-0003-0256-5040
Yahui Yan  http://orcid.org/0000-0001-6934-9874
Matthias P Mayer  http://orcid.org/0000-0002-7859-3112
David Ron  https://orcid.org/0000-0002-3014-5636

## Decision letter and Author response

Decision letter https://doi.org/10.7554/eLife.50793.sa1
Author response https://doi.org/10.7554/eLife.50793.sa2

# Additional files

## Supplementary files

• Transparent reporting form

## Data availability

Diffraction data have been deposited in PDB under the accession code 6SHC. All data generated or analysed during this study are included in the manuscript and supporting files. Source data files have been provided for Figures 1–7.

The following dataset was generated:

| Author(s) | Year | Dataset title | Dataset URL | Database and Identifier |
|---|---|---|---|---|
| Yan Y, Amin-Wetzel N, Ron D | 2019 | Crystal structure of human IRE1 luminal domain Q105C | http://www.rcsb.org/structure/6SHC | RCSB Protein Data Bank, 6SHC |

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
