## [Decision Letter]

**Acceptance summary:**

Eukaryotic cells respond to the accumulation of unfolded proteins in the endoplasmic reticulum (ER) by activating a signal transduction cascade referred to as the Unfolded Protein Response (UPR), which aims to restore ER homeostasis. The IRE1 kinase is an ER-localized transmembrane protein, which serves to detect the accumulation of unfolded proteins and to activate downstream elements of the response. Two predominant, although not mutually exclusive, models have emerged to explain the mechanism by which unfolded proteins lead to IRE1 activation. In one, unfolded proteins bind directly to IRE1's lumenal domain, leading to its activation. In the second model, IRE1's activation by unfolded proteins is more indirect and occurs through their ability to compete for the ER chaperone BiP, which maintains IRE1 in a monomeric, inactive state. Here, by engineering the IRE1 locus to encode a fusion protein in which the effector domain of a BiP co-chaperone (ERdj4) is brought in close proximity to the IRE1 luminal domain, the authors were able to evaluate the consequences of BiP loading onto endogenous IRE1, plausibly without otherwise perturbing conditions in the ER. The dramatic shutdown of IRE1 activity brought about by this manipulation and its relief by inactivating BiP, together provide compelling evidence that BiP can repress IRE1 in cells directly. Further evidence that BiP contributes to IRE1 repression directly in cells, was provided by the observation that deletions in the IRE1 luminal domain, that weaken BiP's ability to enforce a monomeric inactive conformation on the protein, in vitro, also deregulate (enhance) IRE1 activity when introduced as mutations into the endogenous IRE1 locus of cells. Whilst the details of BiP's interaction with the IRE1 and the biophysical basis for direct repression are sketched in low resolution, this paper nonetheless advances our understanding of the UPR by providing important evidence for a component of direct repression of IRE1 by BiP in cells.

**Decision letter after peer review:**

Thank you for submitting your article "Unstructured regions in IRE1α specify BiP-mediated destabilisation of the luminal domain dimer and repression of the UPR" for consideration by *eLife*. Your article has been reviewed by three peer reviewers, and the evaluation has been overseen by a Reviewing Editor and Vivek Malhotra as the Senior Editor. The following individuals involved in review of your submission have agreed to reveal their identity: Eelco van Anken (Reviewer #3).

The reviewers have discussed the reviews with one another and the Reviewing Editor has drafted this decision to help you prepare a revised submission.

Summary:

While all three reviewers found the manuscript to be technically well-executed, there were concerns about the fact that it does not significantly advance the debate on whether BiP binding is critical to Ire1 regulation, if peptide binding is the more critical aspect of activation, or if some combination of the two more closely describes requirements for activation. Also, none of the reviewers were convinced that the data presented in Figure 2A, which demonstrated that the peptides did not significantly alter the stability of Ire1LD, argued against the peptide binding model. But, we believe your findings are important for the field and will appeal to the readers of life. We will therefore gladly consider a revised manuscript provided you can address the essential revisions listed below.

Essential revisions:

1) The discussion should provide a more balanced comparison of the two models for Ire1 activation.

2) The conclusions drawn from Figure 2A should be moderated, unless you are willing to demonstrate the peptides are not inducing dimerization by AUC or a similar type assay.

3) In addition, we also request that you attempt to identify a BiP binding sequence in the unstructured loop that is not present in the structure.

---

## [Author Response]

Essential revisions:1) The discussion should provide a more balanced comparison of the two models for Ire1 activation.2) The conclusions drawn from Figure 2A should be moderated, unless you are willing to demonstrate the peptides are not inducing dimerization by AUC or a similar type assay.3) In addition, we also request that you attempt to identify a BiP binding sequence in the unstructured loop that is not present in the structure.

By way of overview, we note that critiques fell into two categories. The first relate to the experiments dealing with BiP’s role as an IRE1 repressor. The second relate to the insights we bring to the question of the role of peptide binding to IRE1 activation.

In regards to the first issue, Essential revision point 3 charged us with “trying to identify a BiP-binding sequence in the unstructured loop”, whose deletion, we have found, contributes to IRE1 de-repression in cells.

Hsp70 chaperones engage their clients via short extended peptide sequences that interact with the chaperone’s substrate binding domain, an interaction that is greatly stabilised in the chaperone’s ADP-bound conformation. However, the intrinsic affinity of client-derived peptide sequences for the ADP-bound chaperone is but one factor that influences the formation of an Hsp70-client complex. Accessibility of the peptide in the context of the client’s structure and the productive engagement of J-domain protein(s) to promote ATP hydrolysis in a favourable geometry are also important contributors. It is the sum of these components that is relevant to the interaction in vivo. Notably, in this regard, an assay containing IRE1^LD^, a J-domain, BiP and ATP, reveals a role for the flexible region of IRE1^LD^ in the strength of the interaction (Figure 5A). Thus, the correlation between a defect in BiP recruitment to the mutant IRE1^LD ∆∆^ and the de-repressed phenotype observed in IRE1^LD ∆∆^ expressing cells is established. Essential revision point 3, asks if this defect also correlates with an affinity of the flexible region of IRE1^LD^ for BiP in its ADP state.

We began by making use of the scoring system(Blond-Elguindi et al., 1993) that predicts potential BiP binding sites within polypeptides. Applying this algorithm to the entire human IRE1^LD^’s sequence revealed that the highest score (and therefore the highest predicted BiP binding) is represented by the hepta-peptide 309-315 (Author response image 1). The IRE1^LD^ loop deletion altered this peptide from RGSTLPL, with a score of 20, to RGSTDVK, with a score of -9. If this predicted high affinity peptide accounts for the contribution of the flexible loop to BiP binding in vivo, replacing the wild-type hepta-peptide with a low scoring hepta-peptide (in the context of the otherwise intact IRE1^LD^) should de-repress IRE1 signalling.

We tested this by homologous recombination, creating IRE1 variants encoded by mutant alleles of the endogenous Ern1 gene in the dual UPR reporter CHO cell line [converting the wild-type 309-315 hepta-peptide, RGSTLPL (score = 20) either to RGSTDVK (score = -9) or RGSTLSA (score = -12)].

**Author response image 1. respfig1:** BiP binding score based on Blond-Elguindi et al. (1993) of hepta-peptides within the entire human IRE1LD’s sequence. Hepta-peptides having a score of >10 are predicted to have an extremely high probability of binding (black dashed line), peptides with scores between +6 and +10 are predicted to have odds of three to one of having binding activity (grey dashed line), scores between 0 and +5 are predicted to have little predictive value and peptides with scores below 0 are predicted to almost always lack the capacity to bind BiP.

Flow cytometry analysis showed that reconstitution of the locus with either of these mutants resulted in only slightly higher basal IRE1 activity (as reflected in XBP1::Turquoise reporter levels) compared to cells whose locus was repaired with the wild-type sequence. Thus, the highest scoring peptide of the entire IRE1^LD^, appears to account for only some of the repressive effect of the flexible regions of the molecule (Author response image 2).

**Author response image 2. respfig2:** Bar diagram of median XBP1::Turquoise and CHOP::GFP signals from untreated and tunicamycin (Tm)-treated CHO-K1 dual UPR reporter cells with Ern1 alleles encoding wild type (wt) or the indicated mutant variants of IRE1 (∆loop, missing residues 313-338). Data from two independent experiments is shown (means ± range)..

Inability of mutations in the 309-315 peptide to mimic the full effect of the flexible loop deletion may reflect redundancy in the presence of BiP binding peptides (there are five other hepta-peptides with a score between 6 and 10 in the loop, Author response image 1). Such redundancy is also consistent with hydrogen-^1^H/^2^H-exchange mass spectrometry (HX-MS) data which do not point to a prominent, BiP, ATP, ERdj4-dependent protection of any single IRE1^LD^ peptic fragment. Together, these findings suggested that pursuing the in vitro binding characteristics of individual peptides derived from the flexible regions of IRE1^LD^, might not be informative. However, we felt it would be helpful to explore a related question, implicitly raised by the reviewers: “Does the flexible region of IRE1^LD^ make any measurable contribution to the affinity of BiP-ADP to the IRE1^LD^?”

To address this question, we set up an experiment in which BiP was combined with C-terminally biotinylated IRE1^LD^ (either IRE1^LD^-bio or IRE1^LD ∆∆^-bio) in presence of ADP and absence of J-domain protein. Given the slow association of BiP-ADP with substrates and the slow dissociation of BiP oligomers (in the presence of ADP, Preissler et al., 2015), a lengthy equilibration (16 hours, at 4 °C) was allowed. 2.9-fold less BiP was recovered in complex with IRE1^LD ∆∆^-bio than with IRE1^LD^-bio (new Figure 5B). BiP association was concentration-dependent, destabilised by ATP and was not observed with the substrate-binding mutant BiP^V461F^. Coupling of BiP’s nucleotide binding and substrate binding domains was dispensable for the interaction with IRE1^LD^-bio, as it was also observed with the domain-uncoupled BiP^ADDA^ (Petrova et al., 2008; Preissler et al., 2015). Together, these observations point to a role for the flexible regions of IRE1^LD^ in specifying BiP association as a conventional substrate of this Hsp70.

Though deemed a minor point, reviewer 1 noted that “Figure 1D would benefit from including a lane with ERdj4 present with the J-IRE1^LD^-bio construct, as your model would suggest that its inclusion should not significantly increase BiP binding.”

We thank the reviewer for this suggestion and have repeated the entire experiment (twice) with the missing condition added. As expected, no increase in the amount of BiP recovered in complex with J-IRE1^LD^ was observed by the addition ERdj4 (new Figure 1D).

In regard to the second issue, covered by Essential revision points 1 and 2, we should like to begin by noting that we do not believe, nor do want our readers to believe, that we have refuted a role for unfolded proteins as activating ligands of IRE1. Furthermore, like reviewer 3 (Eelco van Aanken), we too recognise that there is no inherent conflict between BiP’s role as a repressor of IRE1 and the possible role of unfolded proteins as activating ligands. At his suggestion, we have revised the Discussion to state plainly: “IRE1 signalling is triggered by an imbalance between unfolded proteins and BiP. The latter results in more potential ligands for IRE1 and fewer molecules of its client protein-free ATP-bound BiP repressor (Bakunts et al., 2017; Vitale et al., 2019). Thus, the two proposed mechanisms for IRE1 activation could well co-exist. However, our findings do raise questions regarding the strength of the experimental evidence supporting the current ideas how unfolded proteins may serve as activating ligands of IRE1”. We believe that this new narrative adequately addresses Essential revision point 1 “that the discussion should provide a more balanced comparison of the two models for IRE1 activation”.

Essential revision point 2 mandates that “the conclusions drawn from Figure 2A should be moderated, unless you are willing to demonstrate the peptides are not inducing dimerisation by AUC or a similar type assay”.

We thank the reviewers for insisting upon this point. Following their suggestion, we have analysed the effect of MPZ-N on IRE1^LD^ dimerisation by size exclusion chromatography (SEC), exploiting the correlation between the distribution of IRE1^LD^ in its monomeric and dimeric state and its peak elution time. Using fluorescently-labelled IRE1^LD^ we could distinguish between peaks arising from IRE1^LD^ and the MPZ-N-peptide itself, greatly simplifying the interpretation of the SEC trace. No effect of MPZ-N peptide (added at concentrations well above the reported K_1/2 max_ of binding to IRE1^LD^, 16 µM, Karagoz et al., 2017) was noted on the peak elution time of IRE1^LD^-TMR. The latter was introduced into the assay at 500 nM, a concentration near IRE1^LD^’s K_d_ for dimerisation (measured by this assay to be ~800 nM, Figure 5—figure supplement 1D). We reasoned that this concentration is close enough to the concentration required for dimerisation to facilitate the task of any agent capable of promoting dimerisation, yet allows plenty of head room for detecting dimerisation – as reflected by the elution profile of IRE1^LD^-TMR introduced at 2 µM. Despite these measures, no trend towards dimerisation was observed (new Figure 2A and 2B). Together with the differential scanning fluorimetry (DSF, Figure 2—figure supplement 1D) our observations do not support a role for the MPZ-N peptide in influencing IRE1^LD^’s monomer-dimer-equilibrium.

Whilst not deemed an essential point, reviewer 2 questions the nature of the apparent incompatibility between our conclusion that MPZ-N does not specifically bind IRE1^LD^ in the MHC-like groove and the reviewer’s interpretation of the NMR data of Karagoz et al., 2017. The reviewer notes “Karagoz et al. did also do PRE experiments, which are very specific for monitoring distance and allow to distinguish between conformation changes and binding. So it is clear from this that the peptide indeed binds into the groove, and this is incompatible with the interpretation by Amin-Wetzel et al.. These are very hard data, and they are not compatible with the interpretation of the authors”. We welcome the opportunity to try and clarify this point.

As noted by the reviewer, the numerous perturbations to the NMR spectrum arising from the presence of unlabelled peptide may reflect either the consequences of a peptide-IRE1^LD^ interaction or an induced conformational change in the IRE1^LD^ (Karagoz et al., 2017). Therefore, they do not constrain the location of the peptide. However a constraint is provided by the paramagnetic relaxation enhancement (PRE) experiments with IRE1^LD^ and an MPZ-proxyl-labelled peptide (Figure 5 in Karagoz et al., 2017, doi.org/10.7554/eLife.30700.016) to which the reviewer is referring. In PRE experiments, the incorporated spin label causes broadening of isoluecine peaks in a range of 10 to 25 Å or entirely erases them within distances of <10 Å (Gottstein et al., 2012). Therefore, the broadening of peaks observed in the NMR spectra of IRE1^LD^ serves as a distance constraint. The strongest such event positions the labelled Cys5 of the MPZ peptide (LIRY**C**WLRRQAALQRRISAME) within 10 Å of Ile186 (shown in red in Figure 2—figure supplement 2A, with a mesh depicting the outer limits of the location of Cys5). Broadenings of other isoleucine peaks position Cys5 within 25 Å of Ile52, Ile124, Ile128 and Ile263 (coloured yellow in the same).

There are 16 amino acid residues between Cys5 and the C-terminus of the MPZ peptide. If extended, the peptide is free to sample a sphere with a radius of ~48 Å from Cys5. Given that the PRE experiments place Cys5 of the peptide anywhere within 10 Å of IRE1^LD^ Ile186, if extended, the peptide is free to explore the entire space encompassed by the grey mesh in Figure 2—figure supplement 2B. Even if constrained to assume a compressed helical conformation, the peptide could be found anywhere within the yellow mesh.

Therefore, the results of the PRE experiments are also compatible with binding of the peptide outside the MHC-like groove, as suggested by Figure 2 of our manuscript. In fact, our experiments and those of Karagoz et al. are consistent with the MPZ peptide binding in locations where peptides have previously been observed crystallographically on the surface of the related yeast IRE1^LD^ (Credle et al., 2005) or the PERK^LD^ (Wang et al., 2018) (Figure 2—figure supplement 2B, in magenta and cyan, respectively). We conclude that the mode of MPZ peptide binding to IRE1^LD^ remains unknown, that it does not obligatorily involve the MHC-like groove and that even the existence of a unique binding site for the peptide on the surface of IRE1^LD^ remains unproven.

References:

Blond-Elguindi, S., Cwirla, S.E., Dower, W.J., Lipshutz, R.J., Sprang, S.R., Sambrook, J.F., and Gething, M.J. (1993). Affinity panning of a library of peptides displayed on bacteriophages reveals the binding specificity of BiP. Cell *75*, 717-728.

Gottstein, D., Reckel, S., Dotsch, V., and Guntert, P. (2012). Requirements on paramagnetic relaxation enhancement data for membrane protein structure determination by NMR. Structure *20*, 1019-1027.

Petrova, K., Oyadomari, S., Hendershot, L.M., and Ron, D. (2008). Regulated association of misfolded endoplasmic reticulum lumenal proteins with P58/DNAJc3. EMBO J *27*, 2862-2872.

Wang, P., Li, J., Tao, J., and Sha, B. (2018). The luminal domain of the ER stress sensor protein PERK binds misfolded proteins and thereby triggers PERK oligomerization. J. Biol. Chem. *293*, 4110-4121.